# Contrails and Their Impact on Shortwave Radiation and Photovoltaic Power Production - A Regional Model Study

Simon Gruber[1], Simon Unterstrasser[2], Jan Bechtold[3], Heike Vogel[1], Martin Jung[3], Henry Pak[3], and Bernhard Vogel[1]

[1]Karlsruhe Institute of Technology – Institute of Meteorology and Climate Research, Hermann-von-Helmholtz-Platz 1, 76344 Eggenstein-Leopoldshafen, Germany
[2]Deutsches Zentrum für Luft- und Raumfahrt (DLR) – Institut für Physik der Atmosphäre, Oberpfaffenhofen, 82234 Wessling, Germany
[3]Deutsches Zentrum für Luft- und Raumfahrt (DLR) – Institut für Flughafenwesen und Luftverkehr, 51147 Cologne, Germany

*Correspondence to:* S. Gruber (simon.gruber@kit.edu)

**Abstract.** A high resolution regional-scale numerical model was extended by a parameterization that allows for both the generation and the life cycle of contrails and contrail cirrus to be calculated. The life cycle of contrails and contrail cirrus is described by a two-moment cloud microphysical scheme that was extended by a separate contrail ice class for a better representation of the high concentration of small ice crystals that occur in contrails. The basic input data set contains the spatially and temporally highly resolved flight trajectories over Central Europe derived from real time data. The parameterization provides aircraft-dependent source terms for contrail ice mass and number. A case study was performed to investigate the influence of contrails and contrail cirrus on the shortwave radiative fluxes at the earth's surface. Accounting for contrails produced by aircraft enabled the model to simulate high clouds that were otherwise missing on this day. The effect of these extra clouds was to reduce the incoming shortwave radiation at the surface as well as the production of photovoltaic power by up to 10 %.

## 1 Introduction

Contrails consist of ice crystals formed in the exhaust plume of aircraft due to mixing of the hot and humid exhaust with cold environmental air. Contrails form in case the Schmidt-Appleman-Criterion (Schmidt, 1941; Appleman, 1953; Schumann, 1996) is fulfilled, i.e. the ambient temperature is below a threshold of around $-45°$ C. With plume temperatures near $-38°$ C to $-40°$ C, contrail particles, which are formed in the liquid phase initially, freeze homogeneously and quickly to form ice crystals. Those ice crystals grow in air with relative humidity above ice saturation and contrails persist. With such a favorable state of the atmosphere, the originally line-shaped contrails undergo various physical processes at the micro scale, spread by the influence of shear and sedimentation and change their structure and microphysical properties. At some point, contrails are no longer distinguishable from natural cirrus in observations. This type of anthropogenic cloud is then called contrail cirrus (Heymsfield et al., 2010) and can have lifetimes of several hours. In a particular case, an 18-hour old contrail-cirrus could be tracked in a satellite imagery (Minnis et al., 1998). Other examples of long-lifetime contrail observations are summarized by Schumann and Heymsfield (2017).

Contrails influence the radiative budget of the atmosphere in a way that is comparable to that of thin natural cirrus clouds (Sausen et al., 2005). Although a number of observations (e. g. Iwabuchi et al., 2012; Vázquez-Navarro et al., 2015; Schumann et al., 2017) and other modeling studies have been performed, important properties, such as the optical depth or the spatial and temporal extent of occurrence, have not been sufficiently investigated and are also not quantified to a satisfactory

extent (Boucher et al., 2013). The radiative forcing of aged contrails and contrail cirrus is of greater importance than the one originating from young, line-shaped contrails (Stubenrauch and Schumann, 2005; Eleftheratos et al., 2007; Burkhardt and Kärcher, 2011).

Previous model studies primarily used global circulation models extended by parameterizations that are able to simulate line-shaped contrails. Here, the global radiative forcing due to contrails was quantified. The values range from roughly 5

to 20 mWm$^{-2}$ depending on the simulated year and assumed contrail properties (Marquart et al., 2003; Stuber and Forster, 2007; Kärcher et al., 2010) and are consistent with satellite-based estimates by Spangenberg et al. (2013). Taking the larger effect of contrail cirrus into account, a global mean radiative forcing of approximately 38 mWm$^{-2}$ (Burkhardt and Kärcher, 2011) (recently updated to 56 mWm$^{-2}$ by Bock and Burkhardt, 2016b) to 40 to 80 mWm$^{-2}$ (Schumann and Graf, 2013) is found. The major drawbacks of these methods are the coarse resolution of the models in both space and time.

Another class of studies simulates single contrails with high resolution LES[1] or RANS[2] models, hereby focusing on contrail formation (Paoli et al., 2013; Khou et al., 2015), young contrails with ages up to 5 minutes and their interaction with the descending wake vortices (Lewellen and Lewellen, 2001; Unterstrasser, 2014) and the transition into contrail-cirrus over time scales of hours (Unterstrasser and Gierens, 2010a; Lewellen, 2014). Usually, the contrail evolution is studied for a variety of environmental scenarios which helps to single out important process and ambient and aircraft parameters. Parameter studies

allow for investigating the conditions under which contrails are persistent and the manner in which microphysical and optical properties change during transition and decay. Because this method is not applicable for a larger number of contrails and often limited to idealized environmental scenarios, it is not suited to quantify the impact of air traffic on the state of the atmosphere. The spatial scale applied in this study lies between those typically used for large-eddy simulations and global climate models. We use the online coupled regional-scale model system COSMO-ART (Baldauf et al., 2011; Vogel et al., 2009). In this context,

online coupled means that meteorology, chemistry and contrail-related processes are simulated in one model at the same grid and one main time step for integration is used. For this study, the model is extended by introducing a new hydrometeor class for contrail ice crystals. The contrail initialization is based on a parametrization of early contrail properties by Unterstrasser (2016). The prognostic equations for contrail ice are similar to those of the natural cirrus ice class. Despite their similar treatment in terms of model equations, the evolution of the two cloud types can be quite different, as their formation mechanisms

are different. Especially with respect to the spatial scale used in the regional scale COSMO-ART model, the presented study is complementary to the aforementioned approaches and is, to our knowledge, the first study of its kind.

Ice crystals in young contrails are considerably smaller than the ones that form in natural cirrus clouds and occur with substantially higher ice crystal number concentrations (Febvre et al., 2009; Schröder et al., 2000; Voigt et al., 2010). Therefore,

---

[1]Large-Eddy Simulation
[2]Reynolds-Averaged Navier Stokes

the original microphysical scheme was extended by a new hydrometeor class that allows for a separate treatment of the small contrail ice crystals separate from natural ice. This approach allows for the investigation of contrail microphysical properties and their changes during the various stages of development represented in a regional atmospheric model. In contrast to other studies using a two-moment microphysical scheme (Bock and Burkhardt, 2016a), the presented model configuration uses no fractional cloud coverage and no prognostic equations for contrail geometric properties like volume or area which describe the contrail spreading on the subgrid scale. Compared to GCM[3] parameterizations, this omission seems acceptable in a regional model, as the spatial resolution is much higher (horizontal grid size of 2.8 km versus 50 km in a GCM).

Regarding the contrail microphysics and the interaction with the meteorological situation, the entire procedure is online coupled, thus allowing feedback processes between contrails and natural clouds in contrast to other models on a comparable grid scale (Schumann, 2012). There, a mixed Lagrangian-Eulerian approach is used instead of the usual Eulerian treatment. This approach allows covering the scale ranging from thousands of single contrails to multi-year global climate simulations (Schumann, 2012; Schumann et al., 2015; Caiazzo et al., 2017).

One of the key goals of this study is thus to quantify the influence of contrails and contrail cirrus on natural high-level cloudiness. Moreover, the radiative properties of contrails and their local influence on the shortwave radiative fluxes at the surface are examined.

The model uses a diagnostic radiation scheme (Ritter and Geleyn, 1992). Because the description of shortwave optical properties for ice clouds is optimized for various naturally occurring crystal habits (Fu et al., 1998; Key et al., 2002), a separate treatment of contrail ice crystals is introduced here as well.

Another feature of this study is the new and recently compiled data set of flight trajectories. Rather than statistical calculations for globally averaged fuel consumption, the basic data consist of real commercial aircraft waypoint data (flightradar24.com, 2015). Another approach using commercial flight data, to study contrails on a regional scale is described in Duda et al. (2004). Here, a combination of commercial flight data and coincident meteorological satellite remote sensing data was used to perform a case study of a widespread contrail cluster. In the future, further case studies should be performed for which in-situ observations of natural ice clouds and especially contrails are available.

The presented model configuration serves to study microphysical evolution of contrails and contrail cirrus, their influence on natural high-level cloudiness, and their impact on the radiative fluxes on a regional scale and short time periods. This gains importance, e.g., in predicting the energy yield from photovoltaic (PV) systems. During the past decades, the development of alternative, clean energy production was enhanced to counteract global warming and reduce air pollution. Within the scope of methods, one of the most promising sources is solar energy, gained by PV cells. To assure a sustainable supply, the demand of precise prediction of the energy yield from PV systems is desired (Lew and Richard, 2010). Several approaches exist to forecast PV power, such as statistical models, neural networks, remote sensing models and numerical weather prediction models (Inman et al., 2013). Especially PV forecast using numerical weather prediction models is challenged by special weather situations or phenomena that are poorly represented in the models (Köhler et al., 2017). E. g. Rieger et al. (2017) found a large impact of mineral dust due to Saharan dust outbreaks on the solar radiation over Germany. This becomes important, as mineral

---

[3]Global Circulation Model

dust is currently not considered adequately in operational weather forecast. Another phenomenon that is not represented at all in numerical weather prediction, is the influence of aviation.

In section 2, the modification of the microphysical scheme and the radiation scheme are described. Section 3 presents a description of the parameterization providing the source terms for contrail ice. In section 4, the results of a case study and comparison with satellite observations are presented.

## 2   Model Description

In this section, the parameterizations to calculate the microphysical properties of ice crystals and the modifications to represent contrails are presented.

The model system COSMO-ART comprises a detailed treatment of aerosol dynamics and gas-phase chemistry (Vogel et al., 2009). Although most of these features are not used for the simulations shown in this study, large parts of the infrastructure contained in COSMO-ART, e. g. the tracer structure and modules for reading emission data, are adopted for the contrails parameterizations.

In this study, the COSMO-ART model is coupled with a comprehensive two-moment microphysical scheme following Seifert and Beheng (2006). Until now, the scheme contained one cloud ice class (besides other hydrometeor types in warm and mixed-phase clouds) that describes ice crystals in high-level ice clouds. Because ice crystals in freshly formed contrails are considerably smaller than those in natural cirrus, the basic microphysical processes in young contrails are treated in a separate, newly introduced *contrail ice class*. From now on, the original ice cloud class is called *cirrus ice class*. The separate treatment allows simulating local bi-modal size spectra.

Consequently, also in the radiation scheme, contrails are treated separately from the cirrus ice using the new contrail ice class. The applied radiation scheme is described in section 2.2.

Unless otherwise indicated, the parameterized processes for the new contrail ice class are the same as those used for the cirrus ice class.

Similar approaches with a separate contrail ice class using climate models with coarser grid size are described by Burkhardt and Kärcher (2009) and Bock and Burkhardt (2016a).

### 2.1   The Contrail Ice Class

In this section, we first describe shortly the treatment of ice crystals in the cirrus ice class. Hereby, we mainly focus on those aspects that are relevant for understanding contrail-specific modifications in the contrail ice class explained later on. A longer description including various formulae as well as a table showing the different coefficients characterizing both the contrail and the cirrus ice class (Tab. 1) is deferred to the appendix.

In each grid box, the ice mass distribution is assumed to follow a generalized $\Gamma$-distribution $f(m)$ (see Eq. A1). Prognostic equations are solved for the ice crystal number concentration $n$ (zeroth moment of $f(m)$) and the ice crystal mass concentration $q_i$ (first moment of $f(m)$). Written concisely, the prognostic equations for the moments $M^k$ of order $k = 0$ or 1 (see Eq. A2

for a definition) read as

$$\frac{\partial M^k}{\partial t} + \nabla \cdot \left[ \boldsymbol{u} M^k \right] - \nabla \cdot \left[ K_h \nabla M^k \right] + \frac{\partial}{\partial z} \left[ \overline{v}_{sed,k} M^k \right] = S^k \tag{1}$$

where $\boldsymbol{u}$ is the grid scale mean wind, $K_h$ is the turbulent diffusion coefficient and $\overline{v}_{sed,k}$ is the number or mass-weighted sedimentation velocity (see Eq. A7). $S^k$ comprises all source and sink terms like nucleation, deposition/sublimation and ag-
gregation.

The deposition source term is derived from the growth equation of a single ice crystal which is integrated over the whole ice crystal mass spectrum.

The ice crystals have a hexagonal shape and the mass $m$ of a single ice crystal is related to its size $L$ via the mass-size relation

$$m = a_{\text{geo}} L^{b_{\text{geo}}} \tag{2}$$

For $m$ in kg and $L$ in m, the parameter values are $a_{\text{geo,nat}} = 1.59$ and $b_{\text{geo,nat}} = 2.56$ (A. Seifert, personal communication, June, 01, 2017) and are valid in the size range $[L_{\min} = 17.5 \, \mu\text{m}, L_{\max} = 3800 \, \mu\text{m}]$.

The treatment of contrail ice is analogous to that of natural cirrus with only a few modifications. Nucleation is switched off in the contrail ice class. Instead, the generation of contrail ice depends on air traffic and the atmospheric state and is explained in section 3.1.

As mentioned before, most ice crystals in contrails are smaller than in natural cirrus and different values for $a_{\text{geo}}$ and $b_{\text{geo}}$ (values are given in the appendix) assure reasonable aspect ratios, also for small ice crystals down to sizes of $L_{\min} = 1.24 \, \mu\text{m}$. The upper size limit is set to a relatively small value of $L_{\max} = 58 \, \mu\text{m}$. If the mean ice crystal size in a grid box exceeds $L_{\max}$, then the total ice crystal mass and number from such a grid box are transferred from the contrail ice class to the cirrus ice class. This is reasonable, as contrails show distinct bi-modal size spectra with many small ice crystals with sizes around $10 \, \mu\text{m}$ and fewer large ice crystals in the fall streaks (Unterstrasser et al., 2017a; Lewellen et al., 2014). The contrail ice class contains predominantly small ice crystals and the cirrus ice class allows for larger ice crystals that may also stem from aged contrails or contrail fall streaks. One drawback of this approach is that the anthropogenic contribution in the cirrus ice class cannot be directly determined. Instead, we analyze the differences in the cirrus ice class between simulations with and without air traffic. This indirect quantification of the aged contrail contribution could be circumvented by introducing further contrail ice classes and may be implemented in the future.

The sedimentation parameterization and other components are as in the cirrus ice class.

The introduction of a second ice cloud class leads to a more complex behavior as both ice cloud classes are coupled and interact with each other in several ways. They are directly coupled via the collection process (see appendix) and the mass transfer of large contrail ice crystals as described above. Moreover, they interact indirectly via the competition for the available water vapor and possibly via dynamical changes through diabatic processes.

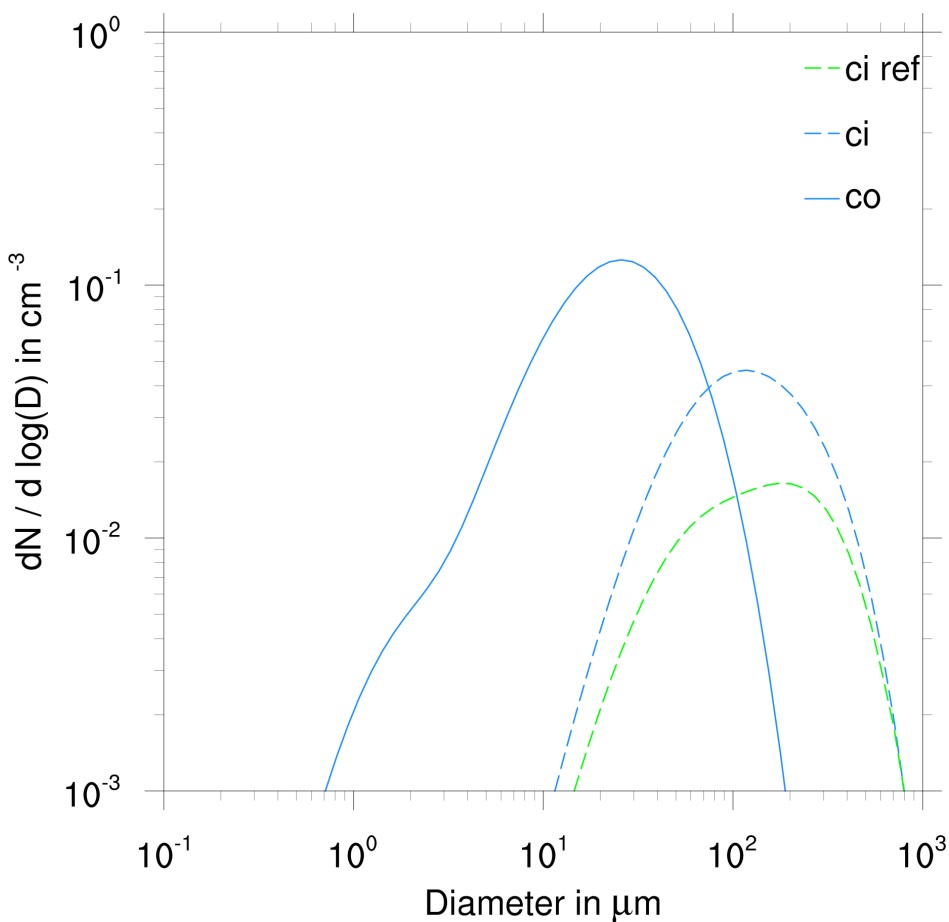

**Figure 1.** Ice crystal size distribution of the simulation with (blue) and without (green) aviation. For the simulation with aviation, separate ASDs of the contrail and cirrus ice class are shown (solid and dotted line). 12 UTC.

For a first illustration of our approach, Fig. 1 shows average size distributions (ASD) of a simulation with and without air traffic. Each ASD is a superposition of local $\Gamma$-distributions. In this example, the contrails are at most 4 hours old. More details on the simulation setup follow in section 4.1. The solid blue line shows the contrail ice class which has a peak at sizes around 20 µm. A less pronounced maximum is located at about 2 µm. The contribution of aged contrails becomes apparent by comparing the two dashed lines. Those show the cirrus ice class ASDs of a simulation with air traffic (blue) and without air traffic (green). Apparently, the anthropogenic contribution is substantial, in particular in terms of total number. The ice crystals in aged contrails are on average smaller than in natural cirrus and the peak size of the SD is shifted to a smaller value.

## 2.2 The Radiation Scheme

The atmospheric radiative fluxes in the COSMO-ART model are calculated using the GRAALS (General Radiative Algorithm Adapted to Linear-type Solutions) radiation scheme (Ritter and Geleyn, 1992). The algorithm needs as input several quantities such as temperature, pressure thickness of the model layers, trace gas concentrations, as well as the cloud cover and the mass
mixing ratio for each hydrometeor class considered (Ritter and Geleyn, 1992).

To include contrails and contrail cirrus in the radiative algorithm, we include a contrail ice cloud cover determined from the contrail ice class mass mixing ratio. Grid cells, where the ice mass mixing ratio exceeds $10^{-8}$ kg kg$^{-1}$ are considered to be covered with contrails or contrail cirrus. The same threshold value is used in Seifert and Beheng (2006) for grid-scale natural ice clouds. As mentioned before, the aviation contribution to the natural cirrus ice class can only be determined by comparison
with the reference simulation.

Within the radiative algorithm, optical properties of hydrometeors are calculated. These are the mass specific extinction coefficient, single scattering albedo, asymmetry factor and the forward peak of the phase function. For ice clouds, the parameterizations following Fu et al. (1998) and Key et al. (2002) are used. Because they are optimized for natural ice clouds, the scheme computes reliable values for effective radii $r_e$ between $5$ and $60$ µm (Fu et al., 1998), whereas for ice crystal populations with
radii smaller than $5$ µm, the parameterization is not well defined. Ice crystals in young contrails often have effective radii smaller than $5$ µm. To overcome this problem, we are using the parameterization of Fu et al. (1998) and Key et al. (2002), but we prescribe a lower limit of $5$ µm for the calculation of the optical properties. For $q_i$, no limit is prescribed; instead, the simulated $q_i$ is used to calculate the optical properties of the ice crystals. As the radiation scheme uses $q_i$ and $r_e$ for determining the radiative fluxes, implicitly fewer but larger crystals are assumed here. The extinction due to small ice crystals is expected
to be larger than that for larger ones. Therefore, in our study, the radiative effect of young contrails may be underestimated.

The calculation of the contrail effective radii follows Fu et al. (1998). Here, the ice crystals are assumed to be randomly oriented, hexagonal columns.

$$r_e = \frac{1}{2} \left( \int_0^\infty D^2 L f_L(L)\, dL \right) \left( \int_0^\infty \left( DL + \frac{\sqrt{3}}{4} D^2 \right) f_L(L)\, dL \right)^{-1} \tag{3}$$

$D$ denotes the half width of an ice crystal which is implicitly given by the mass size relation (see Eq. B2). The size dis-
tribution $f_L$ is related to the mass distribution $f(m)$ via the transformation property $f_L(L)dL = f(m(L))dm$. One can show that $f_L(L)$ (= number concentration per size range) follows a generalized $\Gamma$-distribution, if $f(m)$ (= number concentration per mass range) does so and the mass size relation is a power law (see Eqs. B3 and B4).

Other parameterizations exist that can compute reliable values for optical properties of small ice crystals with sizes down to $0.2$ µm (Bi and Yang, 2017). For future studies, using such an approach clearly could overcome the necessity of the threshold
described above.

## 3 Formation of Contrails

For the description of contrails, the first step is to check whether the environmental conditions are favorable for the formation of contrails. Here, the Schmidt-Appleman-Criterion (Schumann, 1996) is used which defines a threshold temperature below which contrail formation occurs.

In the second step, the source term of contrail ice mass and number has to be calculated. The parameterization used to calculate those source terms is described in detail in Unterstrasser (2016).

### 3.1 Initial Values for Contrails

Microphysical properties of aged contrails depend a lot more on the number of ice crystals than on the ice mass after the vortex phase (Unterstrasser and Gierens, 2010b). The initial ice mass is of minor importance, as the later growth of contrail ice crystals and the related ice mass evolution in a persistent contrail is mainly controlled by the ambient water vapor supply. On the other hand, the ice crystal number changes only slowly in a long-living contrail. Hence, its initial value determines the typical ice crystal sizes in the evolving contrail-cirrus (for a given environmentally controlled ice mass), which affects the radiative properties and the sedimentation-related life cycle.

Therefore, it is appropriate to explicitly prescribe an initial ice crystal number concentration instead of an initial ice mass. The presented procedure is applied at each model time step and for each grid cell, given that both an aircraft is present and the Schmidt-Appleman-Criterion is simultaneously fulfilled.

The parameterization provides ice crystal numbers for contrails that are about 5 minutes old. As meteorological input parameters, it requires the temperature $T$ at cruise altitude, the ambient relative humidity with respect to ice $RH_i$ and the Brunt-Väisälä frequency $N_{BV}$. Furthermore, aircraft properties are characterized by the water vapor emission $I_0$, an 'emission' index for ice crystals $EI_{iceno}$ and the wing span $b_{span}$.

We determine $I_0$ for medium fuel flow at cruise conditions as assumed in Unterstrasser and Görsch (2014). Here, a simple parabolic fit for $I_0$ depending on the wing span $b_{span}$ is used (Unterstrasser, 2016). Information on the wing span is available in the flight track data (see section 3.2).

$$I_0 = c_1 \left( \frac{b_{span}}{c_2} \right)^2; \quad c_1 = 0.02 \text{ kg m}^{-1}; \quad c_2 = 80 \text{ m} \tag{4}$$

In this study only the most common JET-A fuel is assumed to be used; therefore $EI_{iceno}$ is set to $2.8 \times 10^{14}$ kg$^{-1}$ following Unterstrasser (2014).

The total number of ice crystals formed in the beginning, $N_0$, is calculated using the following equation:

$$N_0 = \frac{I_0}{EI_{H20}} EI_{iceno} \tag{5}$$

with water vapor emission index $EI_{H20} = 1.25$. Note that $EI_{iceno}$ is not reduced when the ambient temperature is only slightly below threshold temperature, even though in such situations fewer ice crystals would form (Kärcher et al., 2015).

The descending movement of the primary wake of an aircraft causes adiabatic heating within the plume. Due to this, sublimation and loss of ice crystals occurs, even in a supersaturated environment (e.g. Unterstrasser, 2016). As the spatial and temporal resolution of the model is too coarse to simulate these processes, the fraction of ice crystals surviving the vortex phase is parameterized by a loss factor $\lambda_{Ns}$. Details can be found in Unterstrasser (2016). The total number of surviving ice crystals per flight path $N_s$ is calculated with:

$$N_s = N_0 \lambda_{Ns} \tag{6}$$

For the initial ice mass produced, the water vapor emission $I_0$ is used. The values for the produced ice mass and ice crystal number per flight distance are distributed equally on all grid cells, the aircraft passes within a time step:

$$n_{init} = \frac{N_s d}{V_{cell}}; \quad q_{init} = \frac{I_0 d}{V_{cell}} \tag{7}$$

Here, $V_{cell}$ denotes the volume of the grid cells, $d$ is the flight distance within a grid box.

In global models, contrail parameterizations usually contain further prognostic equations that describe in some way the bulk contrail geometry (e.g. fractional cloud coverage or even some measure of contrail depth). In our approach, we assume that contrail ice crystals always populate the whole grid box as the horizontal scale is much smaller than in GCMs. In the vertical direction, it is assumed that ice crystals are distributed over the whole grid layer. Close to the ground, the vertical grid size is about 10 m and increases to 300 m at the tropopause. In supersaturated conditions, contrail depth varies between 100 m and 500 m (mostly depending on aircraft type and stratification) and is similar to the depth of the grid layer. In the horizontal plane, the simplifications could have a larger effect. If few flight routes transect a grid box and the segments are in total $d_{AT} = 10$ km long then this implicitly results in an hypothetical initial contrail width of $\frac{2.8 \times 2.8}{10}$ km $\approx 800$ m. This is larger than what is typically observed for 5 minute old contrails and better fits to 15 minute contrails (Freudenthaler et al., 1995). Hence, disregarding fractional coverage smears out the initialized contrails to some extent.

## 3.2 Determination of Flight Tracks

In contrast to most of the previously mentioned global modeling studies, this study uses a new and recently derived data set. Rather than statistical calculations for globally averaged fuel consumption, or radar data, the basic data consist of traffic waypoint information over a limited area recorded from ADS-B[4] transponders on the plane (flightradar24.com, 2015). The

---

[4]Automatic Dependent Surveillance - Broadcast

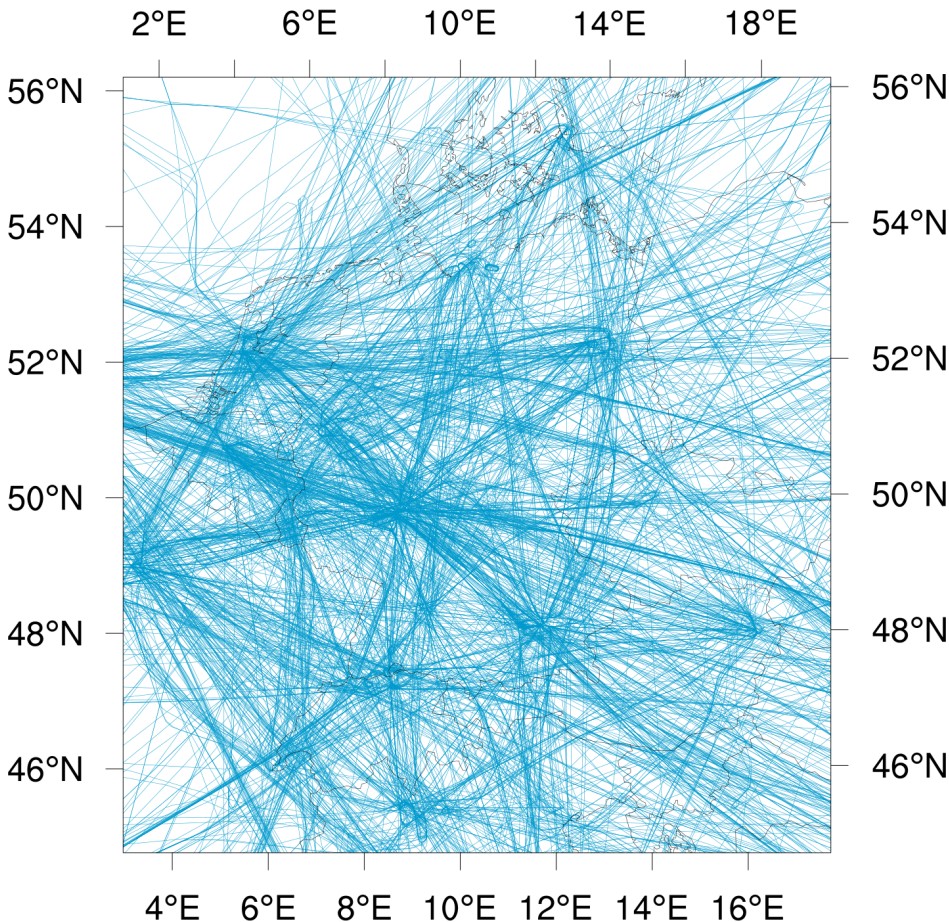

**Figure 2.** Flight trajectories for the simulated domain derived from ADS-B data (08 UTC - 16 UTC 3 December 2013). Each trajectory is plotted as a faint dashed line. Seemingly solid or thick lines are indicative of multiple overlaid trajectories.

ADS-B data is obtained mainly from flightradar24.com (2015). The DLR holds a historic data file that can be purchased from flightradar24.com (2015). This data is cleaned and combined with the input of the Official Airline Guide Database that is also hold by DLR.

From this information, a dataset is compiled containing spatially and temporally resolved information on geographical position, height, current velocity and type of aircrafts. For reasons of efficiency, the information on geographical position is interpolated to fit onto the grid of the model. As a proxy for the aircraft emission parameters (see section 3.1), the wing span $b_{\mathrm{span}}$ is used. The data set contains eight hours of air traffic, beginning at 08 UTC 3 December 2013 and ending at 16 UTC 3 December 2013. The trajectories of all flights during this period are displayed in Fig. 2.

## 4 Case Study

To test the methods described in the previous sections, a case study was performed. For this purpose, a situation over Germany with a high density of contrails in an otherwise cloud-free environment was chosen as the simulation period.

### 4.1 Model Setup

On 3 December 2013, the meteorological conditions over Central Europe were favorable for the formation of contrails. Additionally, the natural high-cloud coverage was relatively low, thus allowing for the identification of contrails on satellite images. For the case study, two simulations were conducted, both running for 24 hours, starting on 3 December 2013, 00 UTC. They use a horizontal $2.8 \times 2.8$ km grid and 60 vertical levels, resulting in a mean distance between the model layers of 300 m in the upper troposphere and 15 m in the lowest model layer. For boundary data, hourly COSMO re-analyses were used. In the reference run, air traffic is turned off and the cirrus ice class is the only ice cloud class to be active. For the run with air traffic, which we call aviation simulation, the previously explained configuration with two ice cloud classes is employed. Air traffic is switched on at 08 UTC and the two simulations evolve identically up to this point. Practically, a spin up phase shorter than 8 h could have been used, but from an operational point of view it was simpler to start the simulations at 00 UTC. As the data set of flight trajectories contains no information about air traffic after 16 UTC, no new contrails form after this time.

### 4.2 Simulated Contrail Properties

The contrail treatment in the microphysics scheme is designed such that contrail-induced changes occur both in the newly introduced contrail ice class and in the existing cirrus ice class. The contribution of young contrails can be directly assessed by evaluating the contrail ice class. The contribution of aged contrails is found by comparing the cirrus ice class of the aviation simulation and of the reference simulation.

Figure 3 shows ice cloud properties over Central Europe in the model layer centered at $z = 11900$ m, at 10 UTC. The various rows (from top to bottom) show the ice water content $IWC$, the ice crystal number concentration $n$, and the effective radius $r_{\mathrm{e}}$. The left column shows the contrail ice class of the aviation simulation, whereas the middle and right columns show the cirrus ice class of the aviation and the reference simulation. At this time, contrails are at most two hours old and mostly consist of numerous very small ice crystals.

Independent of the considered quantity, the contrail ice class features mostly line-shaped structures. The $IWC$ reaches values up to $5 \mathrm{~mg~m^{-3}}$, which is larger than in the simulated natural cirrus. This hints at an accumulation of emitted water vapor additional to the ambient supersaturation. Furthermore, the absolute humidity at the heights considered is relatively low. Therefore, the IWC of natural cirrus is small and cirrus clouds are very thin and almost invisible. The ice crystal number concentrations often lie between $1 \mathrm{~cm^{-3}}$ and $100 \mathrm{~cm^{-3}}$ and can exceed those in natural cirrus by a factor of up to 1000. Consequently, the ice crystal effective radii in contrails are much smaller and lie below $10 \mathrm{~\mu m}$.

The middle panel in Figure 4 shows that the most dense contrails over Northern Germany already leave a mark in the cirrus ice class (see black box in panel b)). Notably, each line-shaped structure in the left panel corresponds to a pair of lines in the

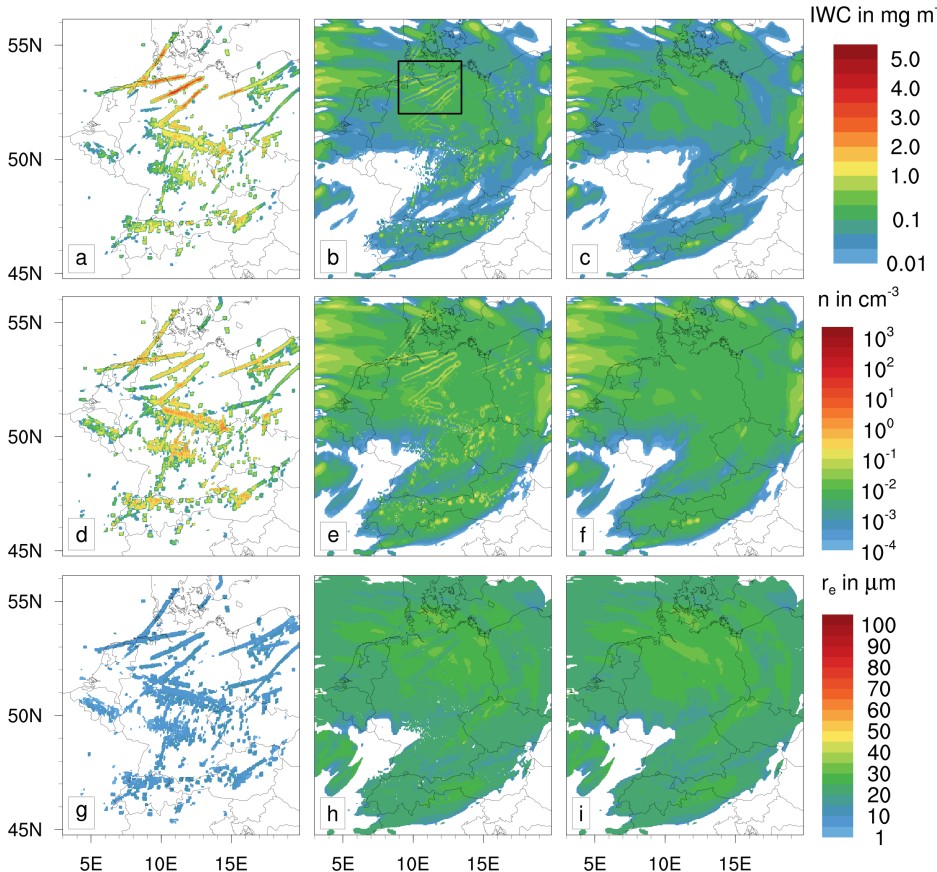

**Figure 3.** Various ice cloud properties are depicted for 10 UTC at an altitude of 11900 m (top row: ice water content; center row: ice crystal number concentration; bottom row: effective radii). Depicted are the simulation with aviation (left column: contrail ice class; middle column: cirrus ice class) and without aviation (right column).

middle panel. This indicates growth of ice crystals particularly at the margins where the contrail ice mass is soon transferred to the cirrus ice class. Consistently, large-eddy simulation studies (Unterstrasser and Gierens, 2010a; Lewellen et al., 2014) and in-situ measurements (Petzold et al., 1997; Heymsfield et al., 1998) indicate the strongest growth at the edges of a contrail. A closer inspection (not shown) reveals that each line-shaped structure in the aforementioned box consists of several contrails.

5    Those were produced by several aircraft that fly along the same route with short time separations.

Nevertheless, the $IWC$, $n$ and $r_e$-values of the aged contrails and the surrounding natural cirrus shown in Fig. 3 are similar. Figure 4 shows the situation at 12 UTC, analogous to Fig. 3 for 10 UTC. As time progresses, existing contrails continue to grow and further contrails are generated. Four hours after air traffic was activated in the model, a major part of the model layer is filled with contrails (left panel). Line-shaped patterns are still identifiable in some places, particularly over Northern

10    Germany. South of $52°$N many contrails overlap and represent a huge contrail cluster. Based on their visual appearance, no clear distinction from natural cirrus would be possible.

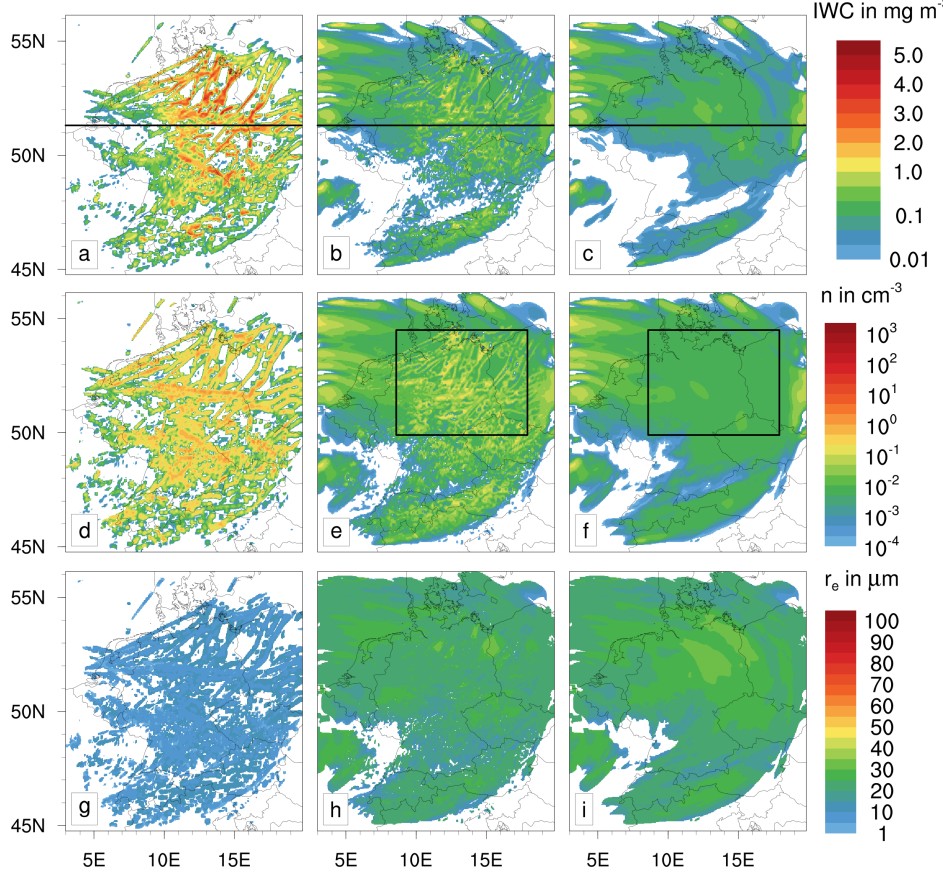

**Figure 4.** Analogous to Fig. 3, now for 12 UTC. The black horizontal lines indicate the location of the curtain displayed in Figs. 5 and 7; black boxes are discussed in the text.

Those contrail clusters still feature high $IWC$, high $n$, and low $r_e$ values comparable to those two hours earlier and distinct to the surrounding natural cirrus. Moreover, we find lots of aged contrails over Germany (black box in middle panel), a region that is basically cirrus-free if there is no air traffic (black box in right panel). Here, local enhancements in both $IWC$ and $n$ occur in the cirrus ice class, where aged contrails are transferred to (Fig. 4e), compared to the reference simulation (Fig. 4f).

The results indicate that, in observations, microphysical criteria may help to separate at least young contrails from natural cirrus. In general, contrail fall streaks and aged contrails cannot be identified as such once their linear shape is lost or masked. Unterstrasser et al. (2017b) shows that natural cirrus that forms in high-updraft scenarios can have ice crystal numbers similar to those of young contrails, which renders even the separation between young contrails and natural cirrus impossible. Moreover, they show that contrails that become embedded in natural cirrus have large volumes where ice crystals of both origins co-exist. Hence, it is no longer meaningful to try to draw a strict separation line between natural cirrus and the anthropogenic cloud contribution.

Next, we analyze vertical distributions along the black line depicted in Fig. 4. Fig. 5 displays the relative humidity $RH_i$ and

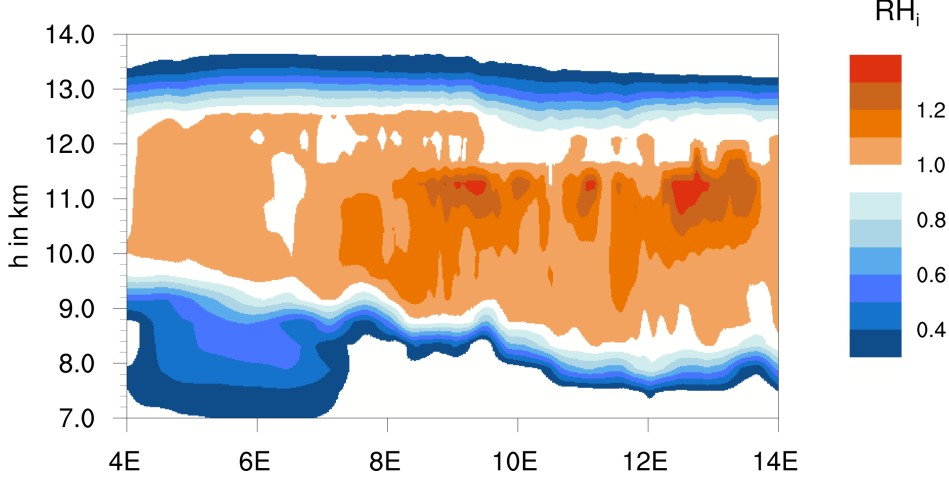

**Figure 5.** Vertical cross section of relative humidity $RH_i$ at 12 UTC along the black line in Fig. 4.

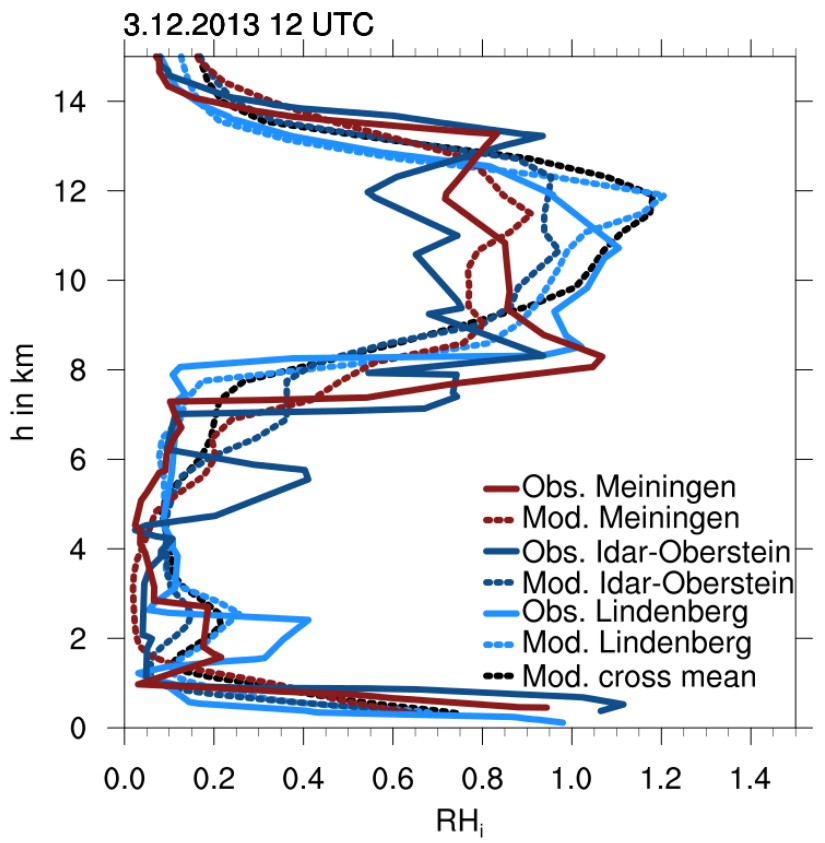

**Figure 6.** Radio soundings for several locations (solid lines) (UWYO, 2018) and corresponding profiles from aviation simulation (dotted lines) of relative humidity $RH_i$. The black dotted line are mean values along the black line in Fig. 4.

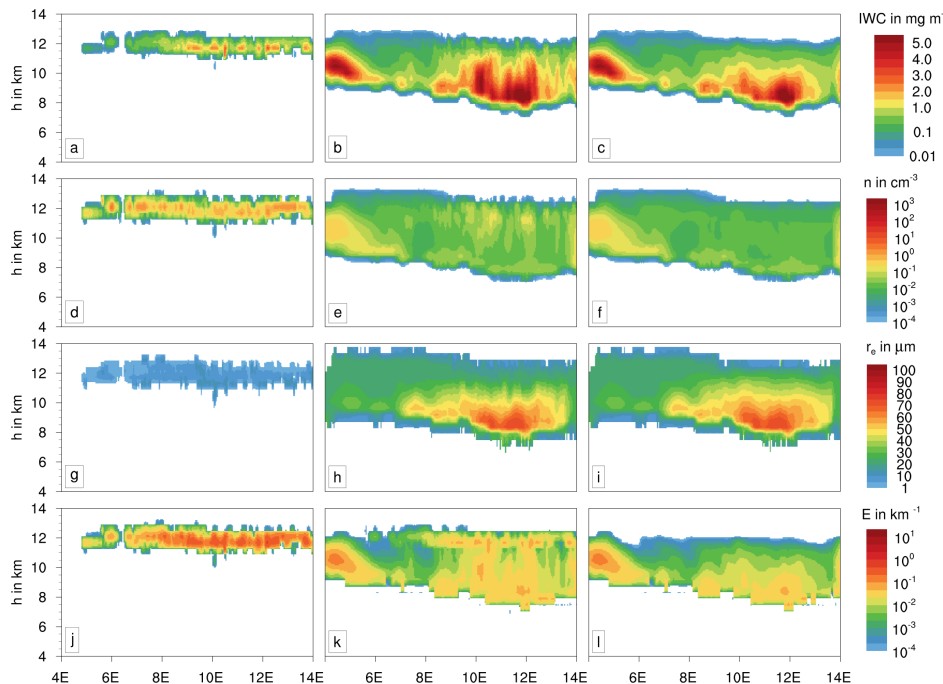

**Figure 7.** Vertical cross section of various contrail properties at 12 UTC along the black line in Fig. 4: ice water content, ice crystal number concentration, effective radius, and extinction coefficient (from top to bottom). Depicted are the simulation with aviation (left column: contrail ice class; middle column: cirrus ice class) and without aviation (right column).

reveals a remarkably thick layer with strong supersaturation (maximum value: 1.34) that extends over the complete east-west extent of the model domain. The layer depth increases from 2.5 km in the west to more than 4 km in the east. These are generally very favorable conditions for the persistence and spreading of contrails.

The fidelity of such high values of $RH_i$ is corroborated by a comparison with observations. Here, vertical profiles obtained from radio soundings (UWYO, 2018) and simulated data evaluated at radiosonde stations are depicted in Fig. 6. In general, high values of $RH_i$ are observed by radio soundings, even supersaturation occurs over Lindenberg and Meiningen. The model clearly overestimates $RH_i$ for Idar-Oberstein, however Meiningen and Lindenberg agree quite well. Additionally, the black curve shows the mean vertical profile of the cross section displayed in Fig. 5. From this it becomes apparent that the relative humidity in the displayed cross section is remarkably high compared to the radiosonde locations. The large layer of supersaturation is caused by lifting and radiative cooling. It can persist, as natural cirrus clouds are located mostly below this layer (Fig. 7b). Also, the cirrus clouds present in the layer are too thin, i. e. occurring number concentrations are too low (Fig. 7e) to effectively reduce supersaturation.

Figure 7 shows the same ice cloud properties as Fig. 4 and additionally the extinction coefficient $E$ at a wavelength of $1.115\,\mu m$. Again, the left column shows the contrail ice class of the aviation simulation, whereas the middle and right column show the cirrus ice class of the aviation and the reference simulation.

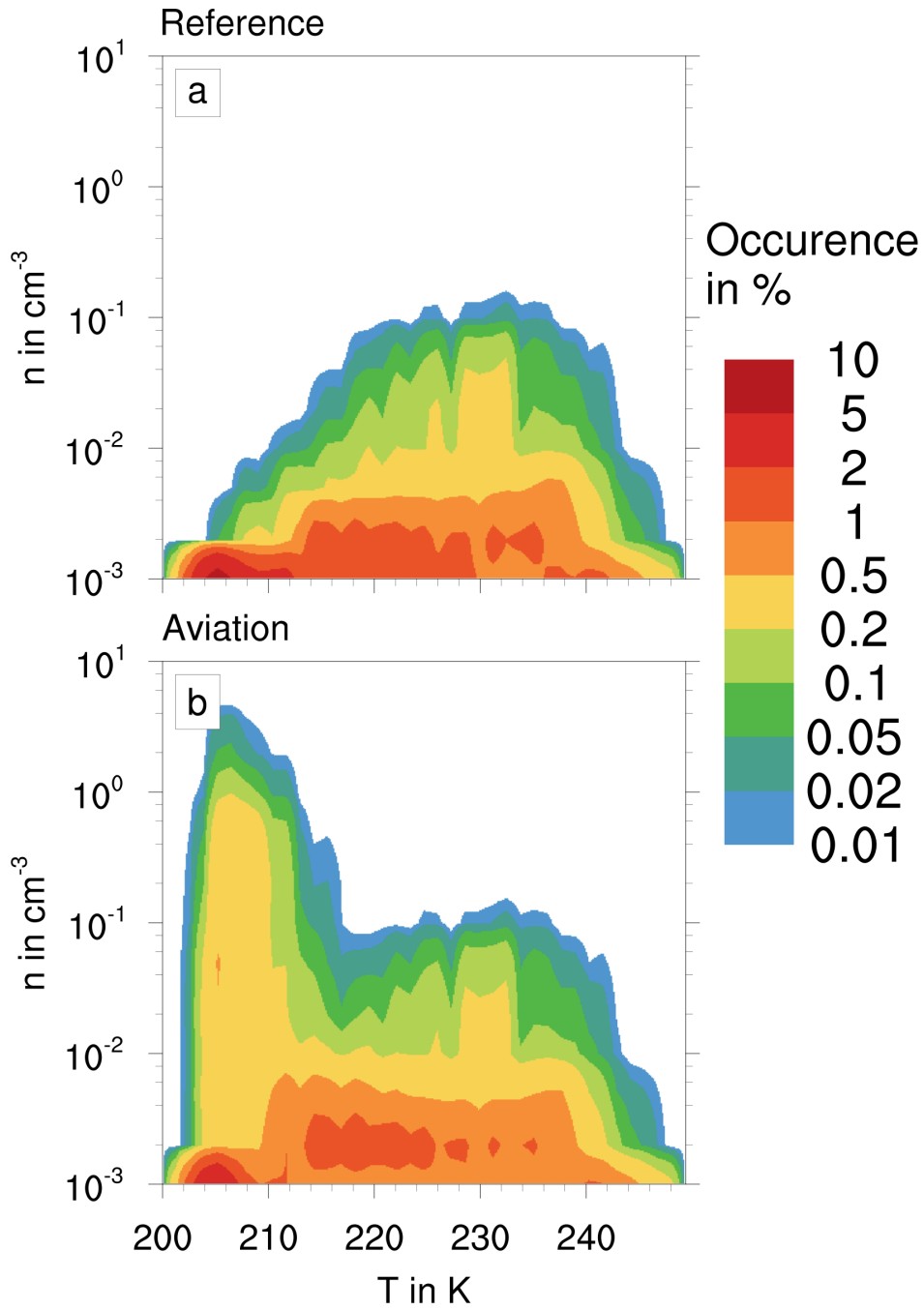

**Figure 8.** Relative occurrence of ice crystal number concentration versus temperature for natural cirrus, contrail and contrail cirrus at 12 UTC for the cross section along the black line in Fig. 4; a) reference simulation; b) aviation simulation.

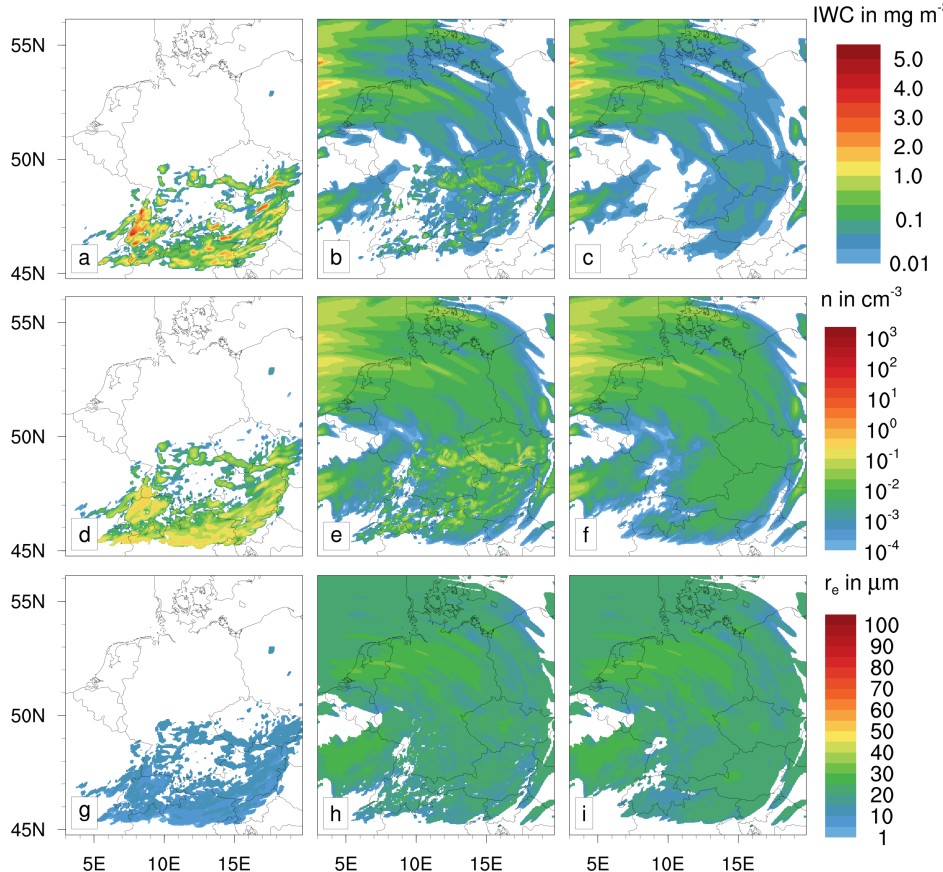

**Figure 9.** Analogous to Fig. 3, now for 20 UTC

In the reference simulation, natural ice is present over the entire supersaturated area. The number concentrations are mostly small, leading to optically thin cirrus clouds with extinction coefficients hardly exceeding $10^{-2}$ to $10^{-1}$ km$^{-1}$. Rather large ice crystals are present throughout the supersaturated area, with the largest values of $r_e$ found in the lower part of the cirrus around 10 to 14° E.

5   In the aviation simulation, it becomes obvious from the left column plots that contrails form only at altitudes between 11 km and 13 km. This is caused by the absence of air traffic below this layer. As already seen in Fig. 4a;b, the $IWC$ is comparable to that of natural cirrus, whereas number concentrations reach much higher values. The effective radii in the aviation simulation are typically one order of magnitude smaller than in natural cirrus clouds and do not exceed 10 µm. These results are in agreement with large-eddy studies (Unterstrasser and Gierens, 2010a; Lewellen et al., 2014) and in situ observations (Poellot et al., 1999). The numerous small crystals lead to high values for the extinction coefficient (up to 2 km$^{-1}$). Therefore, contrail ice is of great importance for the radiation budget.

In Fig. 8, the relative occurrence of ice crystal number concentrations and temperature for the cross section shown in Fig. 7 is

depicted. The relative occurrence is normalized with the sum over all values. Both, reference simulation (Fig. 8a) and aviation simulation (Fig. 8b) are similar for higher temperatures (i. e. lower heights) up to 220 K. For lower temperatures, high number concentrations up to 7 $cm^{-3}$ occur in the aviation simulation, whereas number concentrations clearly decrease strongly with temperature in the reference simulation. Here, a rough comparison to measurement data can be made. In Voigt et al. (2017), mid-latitude cirrus clouds and contrails where probed in-situ during an aircraft measurement campaign. Comparing their Fig. 6(b) to Fig. 8, a similar increase in $n$ below temperatures of about 220 K is found. Therefore, most likely, the high values occurring in the aviation simulation and not in the reference simulation, are due to aviation induced clouds.

In the aviation simulation, changes in the natural ice class can be found mainly at heights where contrails form and slightly below of it. The enhancement of ice number concentrations occurs mostly at flight levels, whereas an increase in $IWC$ is found below. During the initialization, contrail ice crystals are vertically distributed over the whole grid layer and this indirectly accounts for the initial wake vortex induced vertical expansion of a contrail. Within our simulation, the vertical structure of the contrails is determined only due to the gravitational settling of the larger ice crystals. During this process, ice crystals number concentrations tend to decrease. In contrast, only a slight increase in $r_e$ is found. In areas where contrail ice enters the cirrus ice class, a large increase in extinction coefficient occurs. Values of the extinction coefficient are comparable to those of the contrail ice class and can be as high as 1 $km^{-1}$.

After 16 UTC, no new contrails are initialized in our simulation. Four hours after the end of new contrail formation, the remaining contrail ice has been advected to the southern part of domain (Fig. 9). Local patterns of increased number concentrations in the cirrus ice class are now limited to those regions, where contrail ice is still present. The line shaped structures in the contrail ice class vanish, but relatively small values for $r_e$ are still found throughout the domain.

In the northern part of domain, the cirrus ice class is again undisturbed by aviation. Here, no remarkable differences to the reference simulation can be seen.

## 4.3 Comparison with Satellite Observation

In the following, satellite images (created with Global Imagery Browse Services (GIBS) NASA/GSFC/ESDIS, 2018) are shown in Fig. 10 for a qualitative assessment of the simulations. The panels a and b show the "MODIS Terra Corrected Reflectance True Color" at 10 UTC and 12 UTC for the simulated day, respectively, both with a resolution of 250 m. The "True Color" composition consists of MODIS bands 1, 4 and 3 (NASA/GSFC/ESDIS, 2018). Beside a cloud bank over the North and Baltic Seas and fog over Southern and Western Germany, a considerable number of line-shaped contrails and diffuse cirrus clouds are present across both satellite images. Main contrail clusters are found over Central Germany for both situations. Contrails can also be identified over the Netherlands and Belgium, over the Czech Republic, and south of the Alps. At 12 UTC, the high-level cloud cover has increased and pervaded compared to 10 UTC.

Comparing Fig. 10a and Fig. 10e, obviously, the reference simulation underestimates the coverage of high clouds in the center of the domain, which, in large parts, consists of contrails and contrail cirrus (Fig. 10c). Although a number of the patterns of high-level cloud cover detected by MODIS at 10 UTC can be identified in the aviation simulation, the amount of cloud cover seems to be slightly underestimated in Fig. 10c. This discrepancy is probably due to the fact that air traffic was switched on at

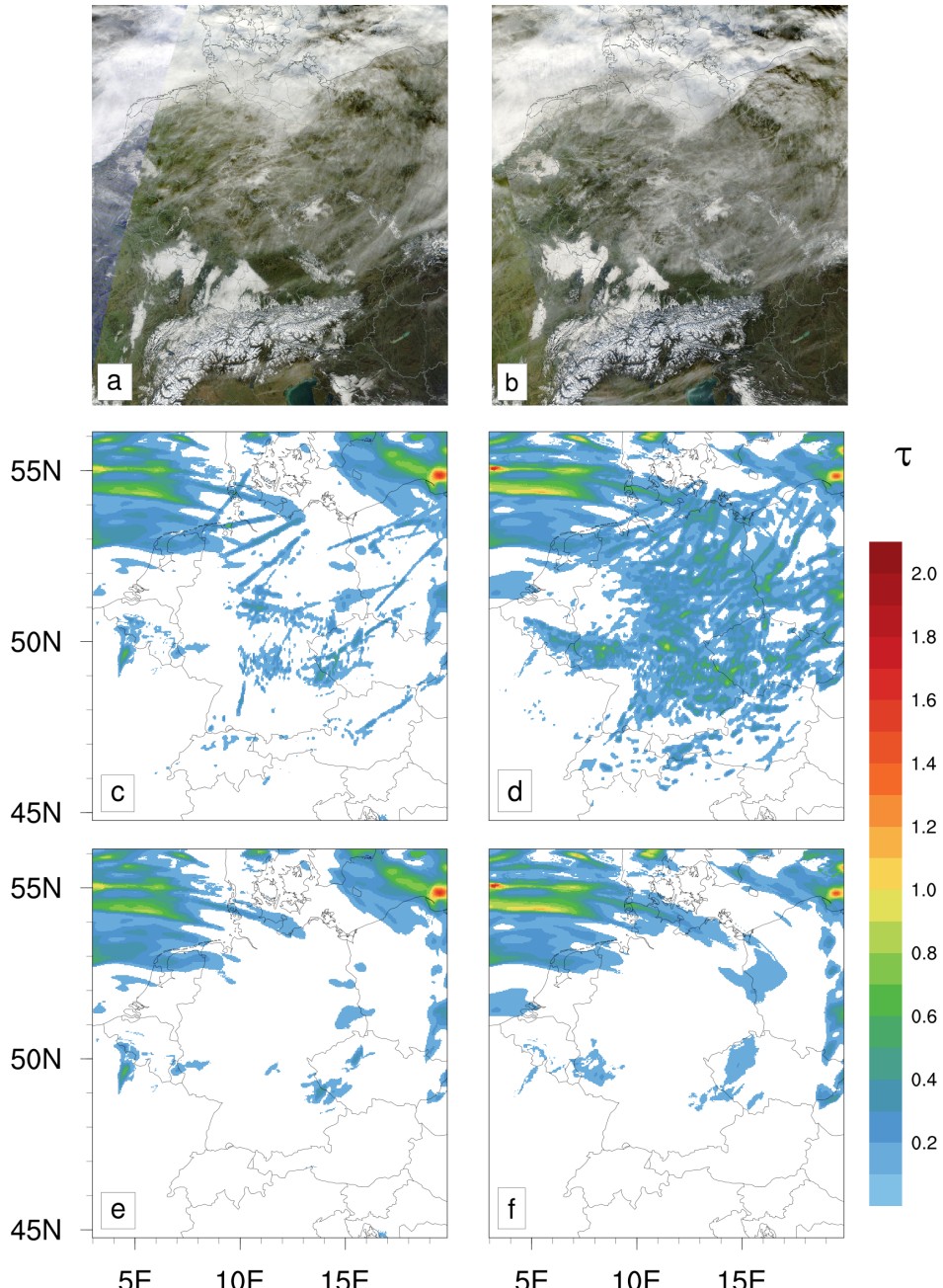

**Figure 10.** Top row: satellite image (MODIS True Color - Corrected Reflectance) (NASA/GSFC/ESDIS, 2018); center row: optical depth at 1.115 μm of all ice clouds for the aviation simulation; bottom row: optical depth of all ice clouds for the reference simulation; left column: 10 UTC; right column: 12 UTC.

08 UTC and earlier flight movements are disregarded in our simulation. Hence, the simulation evaluation at 10 UTC neglects all contrails older than 2 hours. The comparison at 12 UTC is more favorable than the one at 10 UTC. Observations match a lot better with the aviation simulation than with the reference simulation, as in the aviation simulation a much larger fraction of contrails could form since 10 UTC. Mostly over the center of the domain, areas with values of $\tau$ between 0.2 to 0.6 and peaks up to 1.0 are simulated. This is in good agreement with the very high values of the extinction coefficient of contrails compared to those of natural cirrus (Fig. 7j, Fig. 7k, Fig. 7l). The line shaped patterns stem from the contrail ice class and the more patchy structures from aged contrails which have been transferred to the cirrus ice class. The case study for this particular day demonstrates that the inclusion of aircraft effects in a regional weather forecast model improves the representation of high level clouds.

## 4.4 Contrail Impact on Surface Radiative Fluxes

Figure 11 shows the changes in the incoming shortwave radiation (SW) at two points in time at the surface. Clearly, additional ice crystals caused by air traffic reduce the incoming SW radiation. The line-shaped structure of young contrails at 10 UTC (Fig. 3a) is reflected in similar patterns of reduced SW radiation (Fig. 11a). As discussed earlier, the contrail ice class features numerous very small ice crystals with a high extinction coefficient. In the case investigated, this leads to a reduction in incoming SW radiation of 1 to 15 $\mathrm{Wm^{-2}}$. As previously mentioned, our parameterizations are not able to calculate the optical properties of ice crystals with effective radii below the threshold of 5 μm. Rather, crystals with an effective radius of less than 5 μm are assumed to have optical properties of crystals as large as 5 μm. Therefore, the shading effect of the contrail ice crystals might be underestimated in our calculations.

A strong spatial increase in the shading effect is found at 12 UTC (Fig. 11b). Here, the contrail coverage reaches its maximum. Still partly line shaped contrails are found with a similar reducing impact as two hours before. Additionally, clusters of contrail ice transformed into the cirrus ice class have evolved, particularly over the center of the domain (Fig. 4b). The ice crystals herein are relatively small, but larger than those present in the contrail ice class. Therefore, they inhibit less SW radiation from reaching the ground than the ice crystals in the contrail ice class. Consequently, the reduction on SW radiation here is smaller, but still reaches 1 to 10 $\mathrm{Wm^{-2}}$.

Especially in the north of the domain, small areas with the difference in incoming SW radiation attaining large negative values adjacent to large positive values are found. They occur when subtracting e. g. fields of radiative fluxes of the aviation simulation from those of the reference simulation. The introduction of contrail ice acts as source of disturbance for various processes like convection or turbulence. Those features are to be classified as noise as they do not influence the overall situation.

Non-negligible areas with an increase in SW radiation occur, e. g. at 12 UTC over the south-eastern part of the domain (Fig. 11b). Besides only reducing the direct incoming SW radiation, contrail ice crystals also enlarge the flux of diffuse SW radiation (see below). On occasion, as the simulated contrails are still rather optically thin, this effect may be larger than the reduction of direct radiation.

The average change of incoming direct and diffuse SW for 3 December 2013 is shown in Fig. 12. Also here, the small scale fluctuating values in the north are most likely noise.

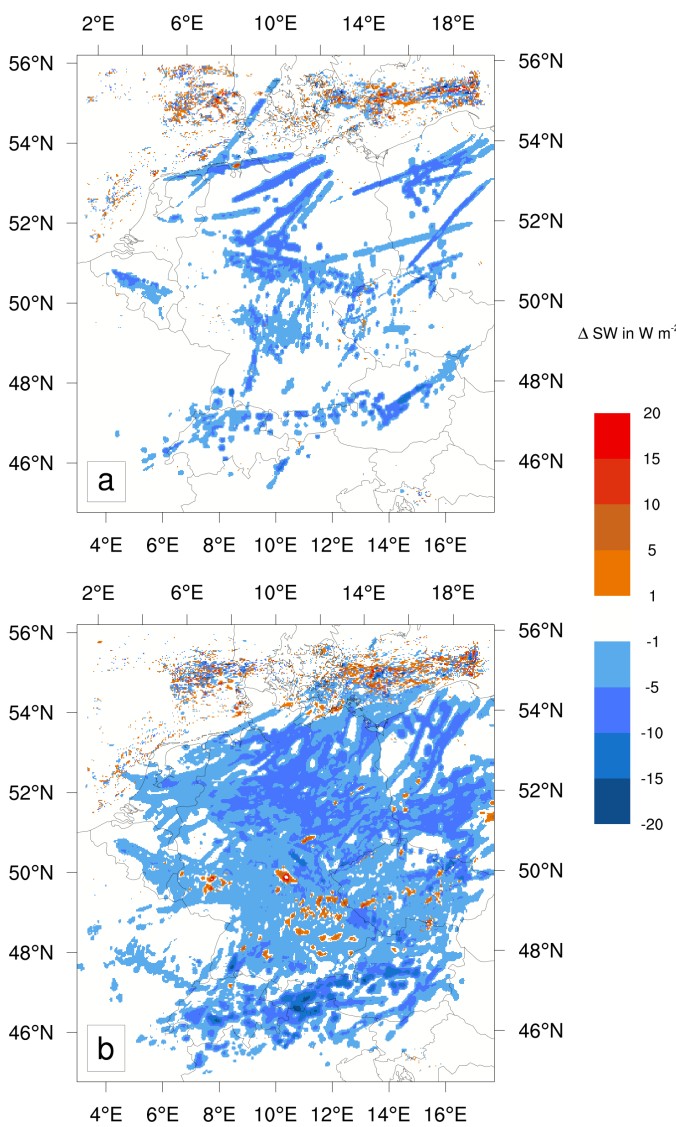

**Figure 11.** Difference in shortwave incoming radiation for aviation simulation and reference simulation for 3 December 2013 at the surface, a) 10 UTC, b) 12 UTC.

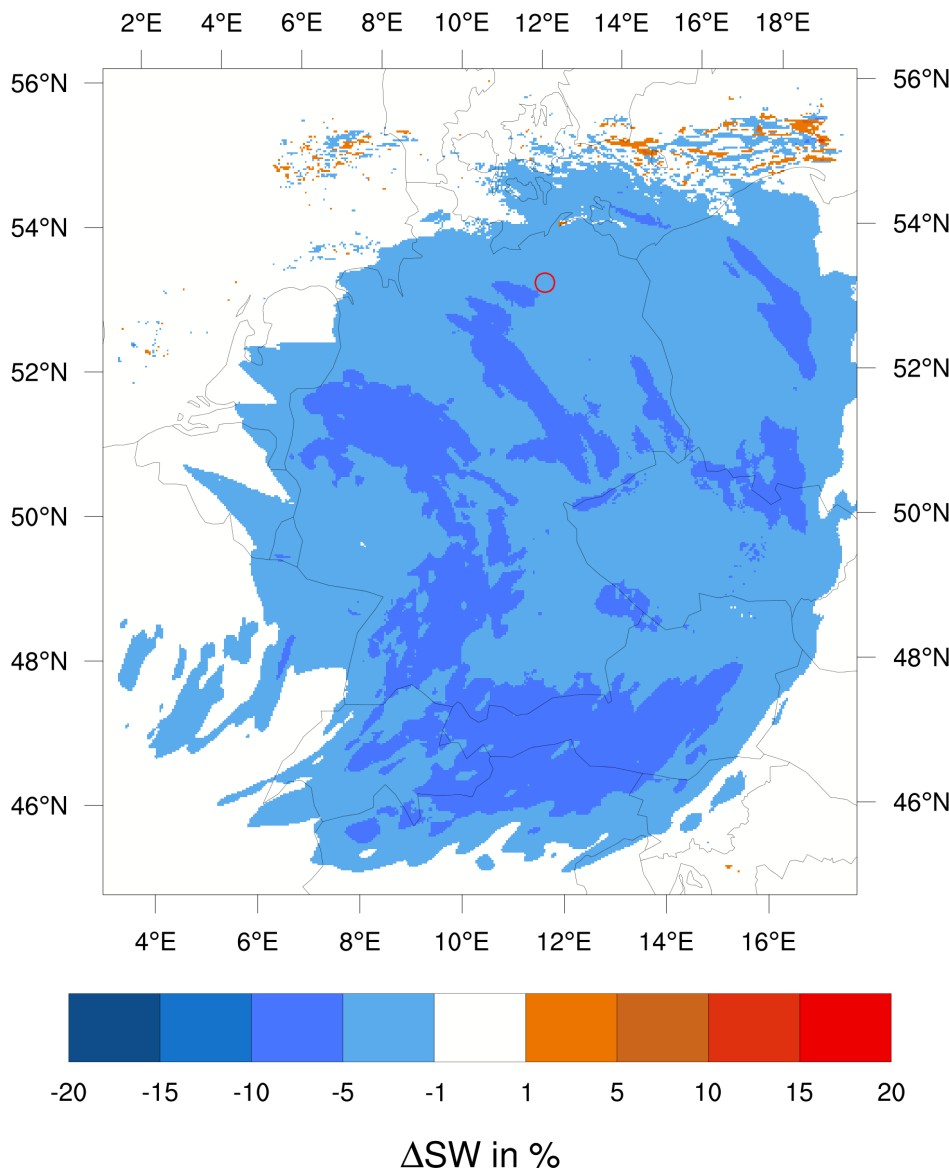

**Figure 12.** Average changes in the amount of incoming diffuse and direct solar radiation during daylight hours of 3 December 2013 at the surface (aviation simulation minus reference simulation). The red circle ($53°$N, $12°$E) marks the location evaluated in Fig. 13.

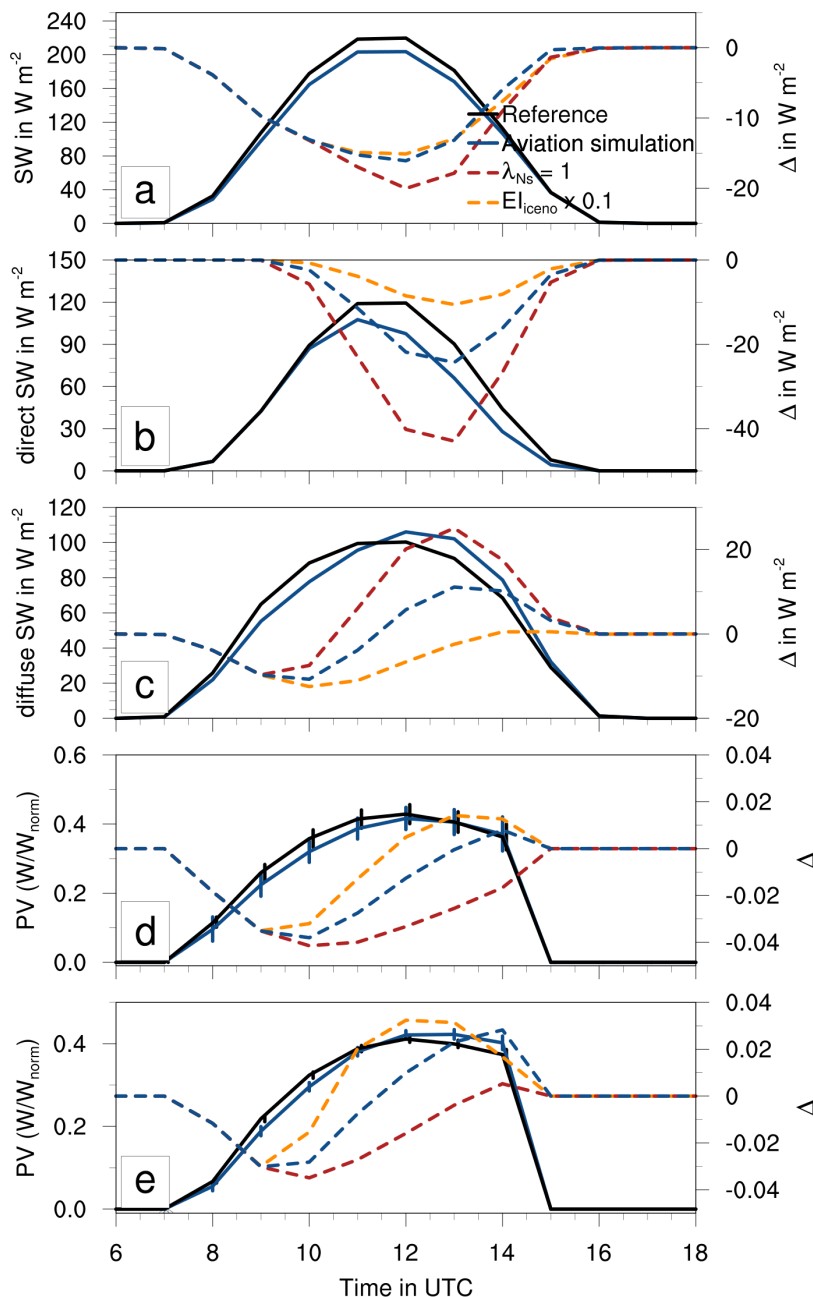

**Figure 13.** Temporal evolution for 3 December 2013 of incoming shortwave radiation: reference (black); aviation simulation as before (blue); "bio-fuel" scenario with $EI_{iceno} \times 0.1$ (orange), omission of crystal loss during contrail vortex phase with $\lambda_{Ns} = 1$ (red). Dashed lines (corresponding to right y-Axis) are difference to reference simulation. a) total SW; b) direct SW; c) diffuse SW; d),e) normalized PV-power. a) - d) over entire domain; e) mean for the location marked with the red circle in Fig. 12. Error bars represent mean values +/- standard deviation.

The thin veil of contrail cirrus that spreads over most of the domain causes an average decrease of incoming direct and diffuse SW radiation of 1 to 5 %. The large and persistent contrail cluster over the northern and eastern part of the domain inhibits on average 5 to 10 % of SW radiation from reaching the ground during approximately eight hours of daylight. The reduction is strongest over the south and the west of the domain. Here, contrail ice is present for the longest time, as seen in Fig. 9.

Fig. 13 shows the temporal evolution of incoming SW radiation and normalized PV power. Panels a to d represent mean values for the entire simulation domain, panel e additionally represents mean values for an area of approximately $50 \text{ km} \times 50 \text{ km}$ centered at a spot in the north-eastern part of the simulated domain, indicated by the red circle (53°N, 12°E) in Fig. 12. This is the location of one of the largest solar parks in Germany (Solarpark Brandenburg-Briest). Also several other major solar parks are located around this area. For enhanced clearness, values for the sensitivity study (see Sec. 4.5) are depicted only

as difference to the reference simulation. The differences between the aviation simulation and the reference simulation are in general more pronounced for the selected location that on average over the entire simulation domain.

During the whole time of daylight, contrails and contrail cirrus reduce up to $15 \text{ Wm}^{-2}$ (about 7 %) of the total incoming SW radiation in the entire domain (Fig. 13a). The effect is largest during noon and ceases during the day. This corresponds to the size of contrail ice crystals. As in our simulation, contrails start to form at 08 UTC, the average contrail ice crystal size grows

during the day. Accordingly, contrail ice effective radii also increase with time and lead to smaller values of the extinction coefficient.

Separating the total SW radiation into its direct (Fig. 13b) and diffuse (Fig. 13c) fraction, it is clear that especially the direct incoming SW radiation is strongly reduced by more than $20 \text{ Wm}^{-2}$ due to the presence of additional ice crystals in the atmosphere. In contrast, the diffuse incoming SW radiation is increased by up to $10 \text{ Wm}^{-2}$. Enhanced scattering of SW radiation

caused by the contrail ice crystals increases the diffuse SW radiation at the ground.

Notably, the peak in reduction of diffuse SW radiation occurs in the afternoon, around 13 UTC. Between 08 UTC and about 11 UTC, the amount of diffuse SW radiation reaching the ground is larger in the aviation simulation. During this time, as mentioned before, contrail ice crystals are smallest on average and forward scattering is less pronounced than for larger crystals, whereas contrail ice crystals grow on average during the day resulting in enhanced forward scattering.

In the aviation simulation, young and aged contrails generally reduce the incoming SW radiation at the surface. This effect is currently neglected in operational weather forecast models. However, this effect is of relevance for the production of solar energy. The temporal evolution of the normalized PV power is depicted in Fig. 13d) and Fig. 13e. The normalized PV is calculated using the open source PV modeling environment PV_LIB for python (Andrews et al., 2014). For a specific combination of a PV module and a PV inverter combination, a nominal power and a reasonable tilt is assumed. These technical specifications are

taken from Rieger et al. (2017). They assume panels consisting of a south-oriented PV module with a nominal power of 220W and a size of $1.7\text{m}^2$. Compared to the reference simulation, the normalized PV power is decreased in the aviation simulation most of the day. The largest losses of up to 10 % occur in the morning and diminish during the day, even an increase occurs for the selected location in the late afternoon (Fig. 13e). The normalized PV power is somewhat more strongly reduced than the total SW radiation. For production of PV power, the incoming direct radiation is of greater importance than the diffuse; of the

two, the direct experiences the larger reduction from contrails and contrail cirrus.

The error bars in Fig. 13d and Fig. 13e represent the mean values +/- the standard deviation with respect to the entire simulation domain and the area of $50 \, \text{km} \times 50 \, \text{km}$ around the selected location, respectively. The standard deviation in Fig. 13d reflects the large-scale variability of the impact of contrails and contrail cirrus on the incoming SW radiation, whereas Fig. 13e illustrates the small-scale variability.

Compared to the selected location, standard deviations are larger for the mean of the domain. Obviously, clouds modify the amount of SW radiation reaching the ground in a non-uniform manner. The magnitudes of the standard deviations for the aviation simulation are about the same magnitude like the ones for the reference simulation. Apparently, the impact of contrails and contrail cirrus on incoming SW radiation is as variable as the impact of natural clouds. E. g., even at 12 UTC, when the impact of contrails and contrail cirrus is largest, confined areas exist, which are unaffected by contrails and contrail cirrus (Fig. 11b).

However, the small-scale variability of the impact of contrails and contrail cirrus on SW radiation is rather small, reflected by much smaller standard deviations in Fig. 13e. One can therefore deduce that the exact location of contrails or contrail cirrus is not crucial for the strength of the impact on SW radiation.

## 4.5   Sensitivity to Initial Ice Crystal Number and Early Contrail Ice Crystal Loss

In this last section, we briefly examine two sensitivities of our model setup.

For the first, we reduce the emission index for ice crystals $EI_{\text{iceno}}$ by a factor of 10 (orange lines in Fig. 13). This scenario explores lower engine soot emissions caused by either improved engine combustor technologies or fuel composition changes from, e. g., biofuel adoption (Moore et al., 2017). Due to this, the initial number concentration of contrail ice crystals is reduced, thus fewer but larger ice crystals are formed (Unterstrasser, 2014). In the simulation with reduced $EI_{\text{iceno}}$, the reduction of total

incoming SW radiation is only slightly weaker than for the simulation assuming standard fuel (Fig. 13a). As ice crystals are slightly larger, their extinction coefficient is lower and the reduction of direct SW radiation is smaller than for the standard setup (Fig. 13b). Also the increase in incoming diffuse SW radiation is less strong compared to the standard setup (Fig. 13c). Consequently, the reduction in normalized PV power is also less strong than for the standard setup, even an enhancement occurs during afternoon (Fig. 13d, Fig. 13e).

Second, we set the surviving fraction of ice crystals $\lambda_{\text{Ns}}$ in the parameterization of the initial ice crystal number to 1. This deliberately neglects the effects of crystal loss during the contrail vortex phase, as parametrized by Unterstrasser (2016). The influence of this parameter is large. During daytime, a reduction in total incoming SW radiation of up to 15 % is simulated (Fig. 13a). Both the reduction in direct SW radiation as well as the increase in diffuse SW radiation are much more pronounced for this case (Fig. 13b, Fig. 13c). Especially the reduction in direct SW radiation causes a strong reduction in production of

PV power. Here, losses of nearly 15 % occur at about 10 UTC. Also concerning the temporal evolution, the reduction lasts much longer compared to the aviation simulation. When no early crystal loss is parametrized in the model, initial ice crystal number concentrations may be much higher than usual. As the initial $IWC$ remains the same, the new crystals are smaller and thus, the simulated contrails are optically thicker. The much stronger reduction in incoming SW radiation demonstrates that

the early ice crystal number loss is non-negligible and an important aspect of contrail evolution as it has a long-lasting impact on contrail-cirrus radiative properties.

## 5  Conclusions

In this study, the regional atmospheric model COSMO-ART coupled with a two-moment microphysical scheme and a diagnostic treatment of radiation was extended by a parameterization describing contrails and the related physical processes. Methods for a separate but consistent treatment of contrail ice were implemented to satisfy the special requirements describing the microphysics in young contrails and the transition phase to contrail cirrus. Performing a case study for a single winter day over Central Europe, it was shown how microphysical properties such as ice water content, ice crystal number concentration and the mean ice crystal radius of ice crystals in contrails change over time and depend on the meteorological conditions.

The ice water content in young contrails is comparable to that in thin cirrus clouds ranging from $0.2$ to $5.0$ $\mathrm{mg\,m^{-3}}$, but with considerably higher ice crystal number concentrations between $1$ $\mathrm{cm^{-3}}$ and $100$ $\mathrm{cm^{-3}}$ and effective radii below $10$ μm. The numerous small ice crystals produce high values for the extinction coefficient and thus also for the optical depth.

The transition of contrail ice into the cirrus ice class causes increasing number concentrations. Here, effective radius of the ice crystals from the contrail ice class grows to an extent comparable to that of natural cirrus. Because of the still relatively high number concentrations, contrail cirrus still features high values for both the extinction coefficient and the optical depth.

Qualitative comparison with satellite data shows good agreement and proves advantages of considering contrails and contrail cirrus in a regional weather forecast model.

Contrail cirrus tends to cause changes in the microphysical appearance of high-level cloud coverage to a remarkable extent, which in turn influences the radiative effect in these regions.

In addition, the impact of contrails and contrail cirrus on shortwave radiation and the production of PV power were quantified. Although the case study was performed for 3 December 2013, when solar zenith angles are low and the length of days is short, a strong influence of contrails is still simulated. They inhibit up to 5 to 10 % of shortwave radiation from reaching the ground at noon. This results in a loss of PV power production of up to 10 %.

Moreover, it was demonstrated that ice crystal loss in young contrails is an important process which can significantly change the contrail-cirrus properties on a regional scale. This study is the first approach to simulate contrails and contrail cirrus using a numerical weather prediction model with high spatial and temporal resolution. Subsequently, the presented method can serve as a basis for improving the predictability of the solar radiation in regional weather forecasting by taking into account contrails and contrail cirrus.

## Appendix A:  Ice microphysical model

This section expands the description of the microphysical model from section 2.1 and presents a collection of underlying equations. All formulae can be found in Seifert and Beheng (2006) as well. Values of constants used in this section are listed

in Tab. 1. The generalized $\Gamma$-distribution is defined as

$$f(m) = A\, m^{\nu} \exp\left(-\lambda m^{\mu}\right) \tag{A1}$$

Here, $m$ is in units of $\mathrm{kg}$. The parameters $\nu$ and $\mu$ are assumed constant, respectively. The parameter $A$ is related to the total ice crystal number concentration $n = M^0$ and $\lambda$ to the ice crystal mean mass $\overline{m} = q_\mathrm{i}/n = M^1/M^0$. Expressions involving $\Gamma$-functions exist for the moments

$$M^k(A, \lambda) = \int\limits_0^\infty m^k f(m; A, \lambda)\, dm \tag{A2}$$

of order $k$ (not shown).

The growth equation of a single ice crystal is given by (Pruppacher and Klett, 1997):

$$\frac{dm}{dt} = \frac{4\pi C(m)\, F_\mathrm{ven}(m)\, S_i}{\frac{R_\mathrm{v} T}{p_\mathrm{sat,i}(T) D_\mathrm{v}} + \frac{L_\mathrm{iv}}{K_\mathrm{T} T}\left(\frac{L_\mathrm{iv}}{R_\mathrm{v} T} - 1\right)} = 4L(m)\, G_\mathrm{iv}(T, p)\, F_\mathrm{ven}(m)\, S_i \tag{A3}$$

Here, $T$ is the temperature, $p$ is the pressure, and $C = D\,/\,\pi$ denotes the capacity of hexagonal crystals (Harrington et al., 1995). $S_i$ is the supersaturation with respect to ice, $L_\mathrm{iv}$ represents the latent heat of sublimation, $p_\mathrm{sat,i}$ denotes the saturation vapor pressure over ice, $R_\mathrm{v}$ is the specific gas constant for water vapor, $K_\mathrm{T}$ is the conductivity of heat, and $D_\mathrm{v}$ is the molecular diffusion coefficient of water vapor. The term $F_\mathrm{ven}$ accounts for ventilation effects and $G_\mathrm{iv}$ considers the diffusion of water vapor and the effect of latent heating:

$$G_\mathrm{iv}(T, p) = \left[\frac{R_\mathrm{v} T}{p_\mathrm{sat,i} D_\mathrm{v}} + \frac{L_\mathrm{iv}}{K_\mathrm{T} T}\left(\frac{L_\mathrm{iv}}{R_\mathrm{v} T} - 1\right)\right]^{-1} \tag{A4}$$

Integration of Eq. A3 over the ice crystal mass spectrum yields the temporal derivative of the ice mass density $q_\mathrm{i}$:

$$\frac{\partial q_\mathrm{i}}{\partial t} = 4 G_\mathrm{iv}(T, p)\, S_i \int\limits_{m_{min}}^{m_{max}} D(m)\, F_\mathrm{ven}(m)\, f(m)\, dm \tag{A5}$$

The mass-size relation given by Eq. 2 uses the values $a_\mathrm{geo,nat} = 1.59$ and $b_\mathrm{geo,nat} = 2.56$ for the cirrus ice class. In order to avoid unreasonably small or large mean masses $\overline{m} = q_\mathrm{i}/n$, a lower limit $m_\mathrm{min} = 10^{-12}$ kg and upper limit $m_\mathrm{max} = 10^{-6}$ kg are introduced. If $\overline{m}$ lies outside the interval $[m_\mathrm{min}, m_\mathrm{max}]$ in a grid box, the ice crystal number concentration is increased to $q_\mathrm{i}/m_\mathrm{min}$ or reduced to $q_\mathrm{i}/m_\mathrm{max}$, respectively. The limits correspond to sizes $L_\mathrm{min} = 17.5$ μm and $L_\mathrm{max} = 3800$ μm. For the contrail ice class, a piecewise definition of $a_\mathrm{geo}$ and $b_\mathrm{geo}$ is employed following (Spichtinger and Gierens, 2009; Heymsfield and Iaquinta, 2000). For masses above $m_\mathrm{split} = 2.15 \times 10^{-13}$ kg (corresponds to $L_\mathrm{split} = 7.42$ μm), $a_\mathrm{geo,con} = 0.04142$ and $b_\mathrm{geo,con} = 2.2$ are used. For masses below $m_\mathrm{split}$, $a_\mathrm{geo,con} = 526.1$ and $b_\mathrm{geo,con} = 3.0$ is prescribed which defines quasi-spherical hexagonal columns

with aspect ratio 1 (see derivation in Spichtinger and Gierens, 2009). The latter constants are valid down to the prescribed lower limit $m_{min} = 10^{-15}$ kg. For grid boxes with lower mean masses, the same bounding technique is used as in the cirrus ice class. The upper limit is set to a relatively small value of $m_{max} = 2 \times 10^{-11}$ kg and the treatment of grid boxes with too large mean masses is different compared to the cirrus ice class. Instead of bounding $n$, the total ice crystal mass and number from such a grid box are transferred from the contrail ice class to the cirrus ice class. The prescribed mass limits of contrail ice class correspond to the size limits $L_{min} = 1.24$ μm and $L_{max} = 58$ μm.

Similar to the mass-size relation, the terminal settling velocity $v$ is approximated by a power law (Eq. A6) with coefficients $a_{vel}$ and $b_{vel}$.

$$v(m) = a_{vel} m^{b_{vel}} \tag{A6}$$

Integrating $v(m)$ and $m \, v(m)$ over the ice crystal mass spectrum yields mean fall velocities $\overline{v}_k$ for the ice crystal number ($k = 0$) and mass ($k = 1$), respectively:

$$\overline{v}_k = a_{vel} \frac{\Gamma\left(\frac{k+\nu+b_{vel}+1}{\mu}\right)}{\Gamma\left(\frac{k+\nu+1}{\mu}\right)} \left[\frac{\Gamma\left(\frac{\nu+1}{\mu}\right)}{\Gamma\left(\frac{\nu+2}{\mu}\right)}\right]^{b_{vel}} \overline{m}^{b_{vel}} \tag{A7}$$

In Eqs. A6 and A7, $m$ and $\overline{m}$ are in kg and $v$ and $\overline{v}_k$ are in in $\mathrm{m\,s}^{-1}$. Using relations relying on power law functions like Eqs. A6 and 2 is beneficial. Then integrals over the mass distribution can often be expressed in terms of moments which avoids employing more expensive numerical quadrature techniques. The contrail ice class uses a piecewise definition of the mass size relation. In this case, truncated moments have to be evaluated.

In the model, both self-collection and collection between the individual classes of frozen hydrometeors are considered. Sink terms (not shown) are included in the prognostic equations of $n$. In case that two different hydrometeor classes are present, where class A has a larger mean mass than class B, the ice crystals of class B are collected by those of class A. In addition to the number loss in class B, mass is transferred from class B to class A.

The nucleation process for natural cirrus is not repeated here as ice crystal generation in the contrail ice class is quite different (see section 3.1)

## Appendix B: Radiation related derivations

For hexagonal columns, the mass of a single ice crystal is given by

$$m = \rho_{ice} \frac{3\sqrt{3}}{8} D^2 L. \tag{B1}$$

Combining the latter equation with Eq. 2 yields

$$D = \sqrt{\frac{8a}{3\sqrt{3}\rho_{\text{ice}}}} L^{\frac{b-1}{2}} \tag{B2}$$

For distributions, the transformation property $f_L(L) = f(m(L))\frac{dm}{dL}$ holds. Using the mass-size relation and the definition of a general $\Gamma$-distribution, it follows with $L$ in units of $\text{m}$:

5 $$f_L(L) = A \left(a\, L^b\right)^\nu \exp\left(-\lambda \left(a\, L^b\right)^\mu\right)\, a\, b\, L^{b-1} \tag{B3}$$

$f_L(L)$ also represents a generalized $\Gamma$-distribution with parameters:

$$A_L = A\, a^{\nu+1}\, b; \quad \nu_L = b(\nu+1) - 1; \quad \lambda_L = \lambda a^\mu; \quad \mu_L = b\mu \tag{B4}$$

Plugging Eq. B2 into Eq. 3 and using the latter relations, the integrals in the effective radius definition can be re-formulated in terms of moments.

10 *Acknowledgements.* We appreciate the constructive comments of all reviewers which helped to improve the manuscript. We acknowledge the use of imagery from the Land Atmosphere Near real-time Capability for EOS (LANCE) system and services from the Global Imagery Browse Services (GIBS), both operated by the NASA/GSFC/Earth Science Data and Information System (ESDIS, http://earthdata.nasa.gov) with funding provided by NASA/HQ.

We acknowledge the use of the radio sounding data provided by the University of Wyoming, Department of Atmospheric Science.

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

**Table 1.** Constants characterizing the natural and the contrail ice class.

| Symbol | Definition | Unit | Natural ice class Value | Reference | Contrail ice class Value | Reference |
|---|---|---|---|---|---|---|
| $a_{geo}$ | parameter in mass-size relation | - | 1.59 | A. Seifert, pers. comm., 01 June, 2017 | 526.1 and 0.04142 | Spichtinger and Gierens (2009), Heymsfield and Iaquinta (2000) |
| $b_{geo}$ | parameter in mass-size relation | - | 2.56 | A. Seifert, pers. comm., 01 June, 2017 | 3 and 2.2 | Spichtinger and Gierens (2009), Heymsfield and Iaquinta (2000) |
| $m_{min}$ | minimum $\overline{m}$ | kg | $10^{-12}$ | Seifert and Beheng (2006) | $10^{-15}$ | this study |
| $m_{max}$ | maximum $\overline{m}$ | kg | $10^{-6}$ | Seifert and Beheng (2006) | $2 \times 10^{-11}$ | this study |
| $L_{min}$ | crystal size corresponding to $m_{min}$ | µm | 17.5 | - | 1.24 | - |
| $L_{max}$ | crystal size corresponding to $m_{max}$ | µm | 3800 | - | 58 | - |
| $\mu$ | parameter of size distribution | - | 0.333 Seifert and Beheng (2006) | | | |
| $\nu$ | parameter of size distribution | - | 1 Seifert and Beheng (2006) | | | |
| $a_{vel}$ | parameter of fallspeed relation | - | 317.0 Seifert and Beheng (2006) | | | |
| $b_{vel}$ | parameter of fallspeed relation | - | 0.363 Seifert and Beheng (2006) | | | |

| Symbol | Definition | Unit |
|--------|------------|------|
| $\Gamma$ | gamma function | - |
| $\lambda, \lambda_L$ | slope parameter of generalized $\Gamma$-distribution | - |
| $\lambda_{\text{Ns}}$ | surviving fraction of ice crystals | - |
| $\mu$ | parameter of generalized $\Gamma$-distribution | - |
| $\nu$ | parameter of generalized $\Gamma$-distribution | - |
| $\rho, \rho_{ice}$ | air density, density of ice | kg m$^{-3}$ |
| $A, A_L$ | scaling parameter of generalized $\Gamma$-distribution | m$^{-3}$ kg$^{-1}$, m$^{-4}$ |
| $a_{\text{geo}}, a_{\text{nat}}$ | parameter in mass-size relation | - |
| $a_{\text{vel}}$ | parameter in fallspeed relation | - |
| $b_{\text{con}}, b_{\text{nat}}$ | exponent in mass-size relation | - |
| $b_{\text{vel}}$ | exponent in fallspeed relation | - |
| $c_{\text{p}}$ | specific heat capacity of air | J kg$^{-1}$ K$^{-1}$ |
| $C$ | capacity | - |
| $CTY$ | cloud type | - |
| $d$ | flight distance | m |
| $D_{\text{v}}$ | molecular diffusion coefficient | m |
| $E$ | extinction coefficient | m$^{-1}$ |
| $E_{\text{AB}}$ | collision efficiency for classes A and B | - |
| $EI_{\text{iceno}}$ | emission index for ice crystals | kg$^{-1}$ |
| $f, f_L$ | number concentration size distribution | m$^{-3}$ kg$^{-1}$, m$^{-3}$ m$^{-1}$ |
| $F_{\text{ven}}$ | ventilation coefficient | - |
| $I$ | ice mass mixing ratio | kg kg$^{-1}$ |
| $q_{\text{init}}$ | "emitted" ice crystal mass concentration | kg m$^{-3}$ |
| $I_0$ | water vapor emission per flight distance | kg m$^{-1}$ |
| $q_{\text{i}}$ | ice crystal mass concentration | kg m$^{-3}$ |
| $K_{\text{T}}$ | conductivity of heat | J m$^{-1}$s$^{-1}$K$^{-1}$ |
| $L$ | ice crystal size | m |
| $L_{\text{iv}}$ | latent heat for sublimation | J kg$^{-1}$ |
| $L_{\text{max}}$ | maximum particle length | m |
| $L_{\text{min}}$ | minimum particle length | m |
| $M^k$ | $k^{\text{th}}$ power moment of $f(x)$ | kg$^k$ m$^{-3}$ |
| $n$ | ice crystal number concentration | m$^{-3}$ |
| $N$ | number concentration | kg$^{-1}$ |
| $N_{\text{BV}}$ | Brunt-Väisälä frequency | s$^{-1}$ |

| | | |
|---|---|---|
| $n_{\text{init}}$ | "emitted" ice crystal number concentration | $m^{-3}$ |
| $N_{\text{s}}$ | number of surviving ice crystals per flight distance | $m^{-1}$ |
| $N_0$ | number of produced ice crystals per flight distance | $m^{-1}$ |
| $p$ | pressure | hPa |
| $p_{\text{sat}}$ | saturation pressure | hPa |
| $r$ | radius | $\mu$ m |
| $r_{\text{e}}$ | effective radius | $\mu$ m |
| $R_{\text{d}}$ | specific gas constant for dry air | J kg$^{-1}$ K$^{-1}$ |
| $R_{\text{v}}$ | specific gas constant for water vapor | J kg$^{-1}$ K$^{-1}$ |
| $RH_{\text{i}}$ | relative humidity with respect to ice | - |
| $S_{\text{i}}$ | supersaturation with respect to ice | - |
| $t$ | time | s |
| $T$ | temperature | kg |
| $v$ | velocity | m s$^{-1}$ |
| $V$ | volume | m$^3$ |
| $b_{\text{span}}$ | wing span | m |