# Peer review of "Contrails and Their Impact on Shortwave Radiation and Photovoltaic Power Production - A Regional Model Study"

_Atmospheric Chemistry and Physics, 2017_

## Referee Comment (RC1) · U. Schumann (Referee) · 14 Nov 2017

This is a nice model study of contrail and contrail cirrus formation on a regional scale. The approach is straightforward in extending a two-moment cloud microphysics scheme by including a separate contrail ice class. The scale jump from young contrails at aircraft wake vortex scales to grid scales is approximated by assuming that the contrail ice spreads immediately over a grid scale (vertically and horizontally). Part of the effects of this strong simplification is corrected by using an ice-crystal loss parametrization derived from LES results.

Of course this simple approach is possible only because the model study is restricted to

regional scales, with just one day of simulation (actually only 8 h of contrail formation). The model is indeed useful to study regional effects of contrail cirrus on shortwave surface radiation and possibly on weather prediction. If applied for climate studies at global scales and for long simulation periods (years), the present approach would suffer from the same problems as other global models. For example, a regional model would require contrail cirrus boundary conditions if run for longer time periods, which are nontrivial if contrail effects outside the region impact the meteorology at inflow. Any climate simulation would also require coupling to oceans etc. Nevertheless, the approach is useful for regional studies and interesting.

Of course, finally, the study should be published, though several minor and some major text issues need to be considered, as listed below, before the paper is acceptable.

Page 1, Line 11 insert "and humid" after "hot".

Line 13 Note that the threshold temperature is pressure dependent; hence, -45°C is perhaps even too rough. More relevant is the fact that below a temperature near -38°C to -40°C, contrail particles, which are formed in liquid phase initially, freeze homogeneously and quickly to form ice particles which then persist in air with relative humidity below liquid saturation (but above ice saturation).

Line 17: there are other long-life-time contrail observations, but I agree, the one described by Minnis et al. (1998) is an early example. Others were summarized in Schumann and Heymsfield (2017), who also review the definition of contrail cirrus and other related knowledge.

Line 20: It is not clear whenever important properties are "sufficiently Investigated", and there are many more observation (see Iwabuchi et al., 2012; Vazquez-Navarro et al., 2015; Schumann et al., 2017) and model studies than those covered in the IPCC report (Boucher et al., 2013).

Page 2, line 17-18: "whether this study is the first of its kind" is at least debatable. In

particular since the authors do not discuss earlier attempts like those of Duda et al. (2004) at this place. But I agree that the simplicity of the present approach including a balanced microphysics model (similar to recent approaches in global models) is attractive and the authors can claim a fresh approach.

Page 3, line 1-3: The method developed by Schumann (2012) and Schumann et al. (2015) (added reference, see below), though certainly with limitations, is still likely the only one covering the scale transition from thousands of single contrails to multi-year global climate cases. This method is not represented fairly in this citation (and in this Introduction) which correctly mentions a problem but misses to mention the advantages of that approach. In fact, the mixed Lagrangian-Eulerian approach could be listed as a basic alternative to the Eulerian grid scale models. This approach has also been used by Caiazzo et al. (2017).

Page 3, lines 13-15: The paper seems to make a big deal out of using 8 h of traffic movements. There were earlier studies doing far more in that direction (e.g. , Schumann, 2012; Voigt et al., 2017).

The authors did not consider a case for which insitu- and satellite observations and other model studies are available, such as the ML-CIRRUS observations of 10 April 2014 over Germany (Voigt et al., 2017), for which the waypoint-traffic data (partly also from flightradar24.com) are available for about 4 weeks and for nearly the whole of Europe. See, e.g., Fig.4 in Voigt et al. (2017). The existence of such data for future studies should be mentioned.

Perhaps, the Introduction should mention the use of satellite data. But it should mention that there were many studies of satellite data in the past (from polar orbiting and geostationary satellites) and also a large variety of in-situ observations is available. So far, I feel that the discussion in this paper is too much biased to LES results instead.

Page 4, Lines 20 to next page: The text explaining the parameters used in Eq. (2) appears a bit lengthy. I assume it can be reordered and shortened.

Page 5, line 23: reorder "several input" -> "input several"

Page 5, line 28: Is there any physical argument for using this maximum mixing ratio limit value? If not, say that this is arbitrarily taken. How sensitive are the results to this threshold?

Page 5, lines 30 ff. The team around Ping Yang has developed an ice particle parameterization including smaller ice particles in recent years (see Bi and Yang, 2017, e.g.). The existence of such parameterizations should be mentioned and such new parameterizations could be used at least in future studies.

Page 9, line 2: The introduction to this section, with "In contrast to previously mentioned global modeling studies,.." and with "globally averaged fuel consumption…" is no longer true if you mention CoCiP properly.

Page 7, line 4: replace "a potential function" by "a power law" or similar.

Page 7, line 13: "Microphysical properties" – this is a very vague term. What do you mean? If you mean optical extinction, I am not sure that your statement is correct (I expect that extinction and optical depth are equally sensitive to number and mass).

Page 8, line 19: 600 m is a large upper bound which is reached very rarely. To be fair, the lower bound should be correspondingly small (100 m; even smaller values occur for small business jets).

Page 9, line 2 ("In contrast to..) : Note that some previous global model studies used similar data for the whole year of 2006 ("ACCRI" waypoint data). Hence, this is not really a big step forward.

The word "exact" does not fit well to this description. When are data exact? Also "real time-based" data is not really the right term. I would simply say you use traffic waypoint data from transponder data (not radar).

Page 11, line 3: Here and at several later places you could also cite observation results.

Here, e.g., Petzold et al. (1997) and Heymsfield et al. (1999) found strongest growth at the edges of contrails from in-situ measurements.

The discussion of observability of contrail cirrus is not needed in this paper. Otherwise, the statements are controversial and would require more in-depth discussion for completeness. For example, observations discriminate contrails and cirrus also based on the concentration of exhaust trace gases and aerosols and use trajectory analysis, partly in correlation with air traffic and meteorological situation history (Voigt et al., 2017). I suggest reducing this discussion in this paper. It is not needed for this paper.

Page 14, line 1 etc.: I agree that Fig. 5 exhibits a remarkably thick layer with apparently strong supersaturation (what is the maximum RHi value in this figure?). I wonder how this layer developed. Is this the results of initial conditions or the result so vertical lifting or radiative cooling? How realistic is the model result in this respect? As far as I understand, the high humidity coincides with some thin cirrus. How can the humidity persist so long in the presence of cirrus? If there is no cirrus yet then I would have expected some homogeneous ice nucleation at such high humidity values. Therefore: How realistic are the high RHi values? Please explain.

Page 14, line 11-12: Why did contrails form only between 11 km and 13 km altitude? Is this because there was no traffic below and above, or because of drier and warmer air above and below?

Page 15, line 2-4. Again you compare to LES only. You could as well compare to observations. Such data are readily available from Schumann et al. (2017).

If you would have plotted the ice particle concentration per volume along a line though the contrail cirrus clouds, you could have compared to the findings in, e.g., Voigt et al.(2017), Fig. 6.

Page 15, line 16: The last sentence of section 4.2 appears trivial. If there is no traffic, there is no chance to affect cirrus that moves in from upstream. That would be different

if you could show that a cirrus parcel which you follow in a Lagrangian manner and that did contain contrails for some time recovered and approached the properties of natural cirrus in a short time after the period with traffic. I think, you cannot show this from this study. So, this sentence should probably be eliminated.

Section 4.3 is on low technical level. It is not even clear from which satellite and sensor the data are taken. What is spatial and temporal resolution of the data? Which spectral channels are used? How sensitive are the observation results to the processing methods used? This needs improvements.

Page 19, Fig. 11: It took me some time to find the red circle. Please add the coordinates (12°E, 53°N) in the figure caption.

Page 22, line 11: What is a "nominal capacity"?

Page 22: Section 4.5: This section depends on the accuracy of the model used (with about 2 km horizontal and 300 m (or 400 m at the tropopause?) vertical resolution) since the early contrail ice crystal loss certainly depends on the time scale of plume mixing. This should be mentioned.

Page 23 line 10-12. The numbers given depend on temperature. See Fig. 5 in Schumann et al. (2017) and Fig. 6 in Voigt et al. (2017), and many other related studies. In fact, this should have been discussed earlier in the text. In the conclusion, it should be said that the numbers for IWC and other contrail-cirrus properties are valid for the specific meteorological situation considered.

Additional references

Bi, L., and P. Yang: Improved ice particle optical property simulations in the ultraviolet to far-infrared regime, J. Quant. Spectrosc. Radiat. Transf., 189, 228-237, doi: 10.1016/j.jqsrt.2016.12.007, 2017.

Caiazzo, F., Agarwal, A., Speth, R. L., and Barrett, S. R. H.: Impact of biofuels on contrail warming, Environ. Res. Lett., 12, 114013, doi:10.1088/1748-9326/aa893b,

2017.

Duda, D. P., Minnis, P., Nyuyen, L., and Palikonda, R.: A case study of the development of contrail clusters over the Great Lakes, J. Atmos. Sci., 61, 1132-1146, 2004.

Heymsfield, A. J., Lawson, R. P., and Sachse, G. W.: Growth of ice crystals in a precipitating contrail, Geophys. Res. Lett., 25, 1335-1338, DOI: 10.1029/98GL00189, 1998.

Iwabuchi, H., Yang, P., Liou, K. N., and Minnis, P.: Physical and optical properties of persistent contrails: Climatology and interpretation, J. Geophys. Res., 117, D06215, doi:10.1029/2011JD017020, 2012.

Petzold, A., Busen, R., Schröder, F. P., Baumann, R., Kuhn, M., Ström, J., Hagen, D. E., Whitefield, P. D., Baumgardner, D., Arnold, F., Borrmann, S., and Schumann, U.: Near-field measurements on contrail properties from fuels with different sulfur content, J. Geophys. Res., 102, 29867-29880, doi: 10.1029/97JD02209, 1997.

Schumann, U., Penner, J. E., Chen, Y., Zhou, C., and Graf, K.: Dehydration effects from contrails in a coupled contrail-climate model, Atmos. Chem. Phys., 15, 11179-11199, doi:10.5194/acp-15-11179-2015, 2015.

Schumann, U., Baumann, R., Baumgardner, D., Bedka, S. T., Duda, D. P., Freudenthaler, V., Gayet, J.-F., Heymsfield, A. J., Minnis, P., Quante, M., Raschke, E., Schlager, H., Vázquez-Navarro, M., Voigt, C., and Wang, Z.: Properties of individual contrails: A compilation of observations and some comparisons, Atmos. Chem. Phys., 17, 403-438, doi:10.5194/acp-17-403-2017, 2017.

Schumann, U., and Heymsfield, A.: On the lifecycle of individual contrails and contrail cirrus, Meteor. Monogr., 58, 3.1-3.24, doi: 10.1175/AMSMONOGRAPHS-D-16-0005.1, 2017.

Vázquez-Navarro, M., Mannstein, H., and Kox, S.: Contrail life cycle and properties from 1 year of MSG/SEVIRI rapid-scan images, Atmos. Chem. Phys., 15, 8739-8749,

doi:10.5194/acp-15-8739-2015, 2015.

---

## Referee Comment (RC2) · Anonymous Referee #3 · 22 Dec 2017

Review of "Contrails and their impact on shortwave radiation and photovoltaic power production – A regional model study" by Gruber et al.

The authors use a regional scale model and contrail parameterization to simulate contrails and cirrus clouds occurring over central Europe during a single day – December 3rd, 2013. The simulated cloud cover and ice crystal mass mixing ratios are used in an online radiation scheme to understand the impact of aviation on direct and diffuse shortwave radiation reaching the surface, which in turn, affect the production of photovoltaic power. Overall, it is reported that aviation-induced cloudiness reduces PV power production by up to 10%. Assumptions related to the emissions index of ice crystals and crystal loss during the contrail vortex phase significantly alter this estimate. This is an interesting case study, which should be worth publishing in ACP; however, the limited time and spatial coverage of the reported model simulations limit the usefulness of authors' conclusions for broader understanding the relevance of aviation contrail cirrus to solar energy production. It certainly would be nice to see more data points for other spatial locations or time periods (e.g., summer). The following comments must be adequately addressed before I can recommend that this paper be acceptable for publication.

1) Pg. 2, Line 16-17: This sentence is confusing. What is being claimed here – that this is the first time a regional scale model has been applied to study contrails and contrail cirrus? I don't think a statement like this is really necessary, but in any case, please be specific with what is being claimed as novel.

2) Pg. 3, Line 12: What is the state of the art with respect to PV forecasts? There does appear to be some literature on this topic using both NWP and statistical models (e.g., Wan et al., 2015). Please expand this section to discuss current methods and considerations.

3) Pg. 3, Line 14-15 and Pg. 9, Lines 2-5: How are the flight radar data obtained and input into the model? How do these flight tracks compare/interface with the ADS-B data presented in Figure 2? Are these data publicly available, and if so, how can the reader obtain the data?

4) Pg. 3, Line 20: In the following what? This sentence is confusing.

5) Pg. 4, Line 21: How often and under what conditions are these limits actually reached in the simulations?

6) Pg. 5, Line 26-28: How are contrail and contrail cirrus ice distinguished from cirrus ice given the statement on Pg. 5, Line 5 that the cirrus and contrail cirrus ice classes are lumped together?

7) Pg. 7, Line 26: Is there a citation for the assumed El\_iceno?

8) Pg. 8, Line 18: What is the vertical resolution of the model?

9) Pg. 10, Line 30: Is this sentence referring to Figure 3 instead of Figure 4?

10) Pg. 12, Line 3-4: What properties are being referred to here? I certainly wouldn't say that the number concentration in 4e and 4f are similar, and there are also large differences in IWC in 4b and 4c.

11) Pg. 14, Lines 11-12: Why do contrails only form at altitudes between 11 km and 13 km? Is this because air traffic is restricted to these altitudes on this line or are there lower altitude flights but the Schmidt-Appleman criterion is only satisfied at these altitudes?

12) Pg. 15, Line 6: Should the first word be "below"?

13) Pg. 15, Line 10: Does the model account for downward subsidence of the aircraft vortices and plumes or is the vertical structure in the modeled contrails only due to gravitational settling of the larger ice crystals? The enhancements in ice number shown in Figure 6 appear to occur at flight level, but the enhancement in IWC is below flight level.

14) I would like to see the satellite observations more directly integrated into the discussion surrounding Figures 3-4 rather than in its own section since I think that it can provide a lot of context and validation for the model results. Figures 8 and 9 as they stand now are kind of on their own and not particularly informative other than to denote that there are thin, high-level cirrus and no low clouds. The cirrus clouds shown in Figure 8a appear to be much more diffuse than the MODIS imagery for this time period, with the MODIS images showing a lot of contrail structure and providing a good snapshot of the time evolution of the scene during the two simulations. I suggest the authors strike Figures 8 and 9, and add MODIS satellite images at 1000Z and 1150Z either as part of Figures 3 and 4 or as a separate figure before them. Such an example figure created from worldview.earthdata.nasa.gov images is on the next page with detailed web references at the end of this review.

15) What is the coordinate chosen for the red circle in Figure 11 and related timeseries analysis in Figure 12? Why was this coordinate chosen? Do the results change if a set of coordinates in the  $\Delta$ SW

---

## Referee Comment (RC3) · Anonymous Referee #1 · 22 Dec 2017

This paper describes a parametrisation of contrails that is embedded in the two-moment cloud microphysics of the COSMO regional atmospheric model. It is followed by a case study including the impact of contrails and contrails-cirrus on short wave radiation and on possible PV production.

The paper presents interesting new results, but before being published the authors should address the following points.

General remarks:

- I was a bit surprised that the authors have chosen a model with a complete representation of atmospheric chemistry and aerosols (COSMO-ART) for this study. From the

description of the parametrisation in Section 2 it seems to me that changes by aircraft to atmospheric chemistry as well as soot emissions are not directly considered. Thus it seems that the usage of the COSMO model (without the computationally expensive ART modules) in conjunction with the 2-moment scheme of Seifert and Beheng (2006) are sufficient for the present study. If this is not the case it should be shown more clearly why the ART modules are needed.

- Whereas the case study in section 4 is well described, sections 2 and 3 are rather hard to read and understand and would profit from a rewrite. In section 2.1 it is very difficult to understand where the description of the standard cirrus class is ended and the description of the newly introduced contrail class starts. I would thus recommend separating by introducing a new section. Also this section would highly profit from a table explaining the differences between both classes as well as a schematics such as Fig 1 in Salzmann et al. (2010) to highlight the interactions of the newly introduced contrail ice class with the other ice classes. A Table is provided at the end of the paper explaining the differences, but is not referenced in the text. I would moreover suggest in Table 1 to distinguish more clearly between the cirrus and contrail ice class. Similarly in section 2.2 a Table would help to understand the differences between the interactions with radiation of both classes.

- It seems quite strange that a completely new scenario "bio fuels" is introduced in the last section of the paper. but referenced already in Fig 12. I would suggest to keep section 4.5 as sensitivity case study, and not introduce a new scenario here. As shown by Ferrone (2011) biofuels also have an impact on the Appleman-Schmidt criterion and this would need to be changed accordingly.

Specific remarks:

- in the last line of page 2, the resolution of Global Circulation Models (GCMs, the abbreviation is not introduced) are given as 250km, however most recent models have a resolution of 50 km or higher (IPCC, 2015).

- Caption of Figure 2: The abbreviation COSMO-DE is not introduced

- Figures 3, 4, 6 and 7 would become more interesting and easy to interpret if a difference plot between the middle and right column would be added.

- Figure 8 and 9: If I understood correctly, the black boxes should highlight the same areas but they are slightly shifted.

- In the list of abbreviations given on page 31 and 32 some Units are erroneous. Units such as kg^(-/mu), m^(-/mu); kg^(-b_vel); kg ^k do not exists.

References:

Ferrone, A, (2011): Aviation and climate change in Europe : from regional climate modelling to policy-options. http://hdl.handle.net/2078.1/74779

IPCC, 2013: Climate Change 2013: The Physical Science Basis. Contribution of Working Group I to the Fifth Assessment Report of the Intergovernmental Panel on Climate Change [Stocker, T.F., D. Qin, G.-K. Plattner, M. Tignor, S.K. Allen, J. Boschung, A. Nauels, Y. Xia, V. Bex and P.M. Midgley (eds.)]. Cambridge University Press, Cambridge, United Kingdom and New York, NY, USA, 1535 pp, doi:10.1017/CBO9781107415324.

Salzmann, M., Y. Ming, J. C. Golaz, P. A. Ginoux, H. Morrison, A. Gettelman, M. KrÃÂďmer, and L. J. Donner, 2010: Two-moment bulk stratiform cloud microphysics in the GFDL AM3 GCM: Description, evaluation and sensitivity tests. Atmospheric Chemistry and Physics, 10, 8037-8064, doi:10.5194/acpd-10-6375-2010.

Seifert, A. and Beheng, K. D.: A two-moment cloud microphysics parameterization for mixed-phase clouds. Part 1: Model description, Meteorol. Atmos. Phys., 92, 45 – 66, doi:10.1007/s00703-005-0112-4, 2

---

## Author Comment (AC1) · 29 Jan 2018

Dear Mr. Schumann,

We thank a lot for your valuable comments and suggestions. We followed them as explained below.

The reviewers comments are repeated in **bold letters,** our replies are given in *italic,* and text modified or added to the manuscript is given in blue.

**This is a nice model study of contrail and contrail cirrus formation on a regional scale. The approach is straightforward in extending a two-moment cloud microphysics scheme by including a separate contrail ice class. The scale jump from young contrails at aircraft wake vortex scales to grid scales is approximated by assuming that the contrail ice spreads immediately over a grid scale (vertically and horizontally). Part of the effects of this strong simplification is corrected by using an ice-crystal loss parametrization derived from LES results.**

**Of course this simple approach is possible only because the model study is restricted to regional scales, with just one day of simulation (actually only 8 h of contrail formation). The model is indeed useful to study regional effects of contrail cirrus on shortwave surface radiation and possibly on weather prediction. If applied for climate studies at global scales and for long simulation periods (years), the present approach would suffer from the same problems as other global models. For example, a regional model would require contrail cirrus boundary conditions if run for longer time periods, which are nontrivial if contrail effects outside the region impact the meteorology at inflow.**

**Any climate simulation would also require coupling to oceans etc. Nevertheless, the approach is useful for regional studies and interesting.**

**Of course, finally, the study should be published, though several minor and some major text issues need to be considered, as listed below, before the paper is acceptable.**

**The authors present a new contrail cirrus parameterization within the COSMO-ART model and study the impact of contrail cirrus on the SW radiation. The work is different to existing studies since they implemented their parameterization in a weather forecasting model. This would actually enable the authors to compare their results to observations but they do not do that. Instead they discuss the development t of a contrail field in simulations only. Nevertheless, I find the model itself and the impact on the SW radiation interesting. There are a couple of major issues that need clarification and/or sorting out. The paper needs to be expanded regarding a more detailed evaluation with observations and a better comparison with earlier work.**

*Dear Mr. Schumann,*

*Thanks a lot for your interest in our work and your detailed review. The issues raised in this review as well as the many questions and explanations clearly help to improve this manuscript. We hope to have addressed them to a satisfactory extent.*

**1. Page 1, line 11: insert ”and humid” after "hot".**
*Done*

**2. Page 1, line 13: Note that the threshold temperature is pressure dependent; hence, -45°C is perhaps even too rough. More relevant is the fact that below a temperature near -38°C to -40°C, contrail particles, which are formed in liquid phase initially, freeze homogeneously and quickly to form ice particles which then persist in air with relative humidity below liquid saturation (but above ice saturation).**

We change in page 1, line 13:

Following the Schmidt-Appleman-Criterion (Schmidt, 1941; Appleman, 1953; Schumann, 1996), contrails form only when the ambient temperature is below a threshold temperature of roughly -45°C. With a favorable state of the atmosphere, i.e. characterized by supersaturation with respect to ice, the originally line-shaped contrails undergo various physical processes at the micro scale…

*to:*

Contrails form in case, the Schmidt-Appleman-Criterion (Schmidt, 1941; Appleman, 1953; Schumann, 1996) is fulfilled, i.e. the ambient temperature is below a threshold of around -45°C. With plume temperatures near -38°C to -40°C, contrail particles, which are formed in the liquid phase initially, freeze homogeneously and quickly to form ice crystals. Those ice crystals grow in air with relative humidity above ice saturation and contrails persist. With such a favorable state of the atmosphere, the originally line-shaped contrails undergo various physical processes at the micro scale …

**3. Page 1, line 17: There are other long-life-time contrail observations, but I agree, the one described by Minnis et al. (1998) is an early example. Others were summarized in Schumann and Heymsfield (2017), who also review the definition of contrail cirrus and other related knowledge.**
*We add in page 1, line 17:*
Other examples of long-lifetime contrail observation are summarized by Schumann and Heymsfield (2017).

**4. Page 1, line 20: It is not clear whenever important properties are "sufficiently investigated", and there are many more observation (see Iwabuchi et al., 2012; Vazquez-Navarro et al., 2015; Schumann et al., 2017) and model studies than those covered in the IPCC report (Boucher et al., 2013).**
*Even taking into account the newer observations you mention we still believe that our statement is reasonable.*

**5. Page 2, line 17-18: "whether this study is the first of its kind" is at least debatable. In particular since the authors do not discuss earlier attempts like those of Duda et al. (2004) at this place. But I agree that the simplicity of the present approach including a balanced microphysics model (similar to recent approaches in global models) is attractive and the authors can claim a fresh approach.**
*We weaken the statement and mention the study of Duda later in the manuscript as it also uses real flight data. However, we do not consider the contrail advection tool of Duda to be a "full" contrail model, in the sense that ice microphysics is included. Hence, our aforementioned claim is not invalidated by disregarding the Duda work.*
*We add in page 2, line 18:*
Another approach using commercial flight data to study contrails on a regional scale is described in Duda et al. (2004). Here, a combination of commercial flight data and coincident meteorological satellite remote sensing data was used to perform a case study of a widespread contrail cluster.

**Perhaps, the introduction should mention the use of satellite data. But it should mention that there were many studies of satellite data in the past (from polar orbiting and geostationary satellites) and also a large variety of in-situ observations is available. So far, I feel that the discussion in this paper is too much biased to LES results instead.**

*We now cite several more studies of contrail observations.*

*We do not think that the presentation or plausibility issues are overly biased towards LES, but it is clear that LES results build an integral part of the contrail initialisation used in our model.*

*At the time the manuscript was prepared, ML-Cirrus results were not published yet and it is not clear whether we would have been granted access at that phase to all the data needed to make conclusive comparisons.*

*We add the following after page 3, line 15:*

In the future, further case studies should be performed for which in-situ observations of natural ice clouds and especially contrails are available.

[Figure]

*Fig. 1: 2014/04/10: Comparison of COSMO-ART ice crystal number concentration (contour) to SID3 measurements (ML-CIRRUS).*

*We also performed a case study for 10 April, 2014 to compare with ML-CIRRUS data (Fig.1). In the future, we want to extend this work and present it in a separate study.*

**8. Page 4, lines 20 to next page: The text explaining the parameters used in Eq. (2) appears a bit lengthy. I assume it can be reordered and shortened.**

*We agree that the mentioned section is more technical than others. We condensed this part and moved the technical issues to the appendix.*

**9. Page 5, line 23: reorder "several input" -> "input several"**
*Done*

**10. Page 5, line 28: Is there any physical argument for using this maximum mixing ratio limit value? If not, say that this is arbitrarily taken. How sensitive are the results to this threshold?**
*In fact, this threshold is used for grid-scale ice clouds in the radiation scheme. For reasons of consistency (and as we find no other suitable study to justify another number) we leave it as it is. Indeed, the results do not seem to depend strongly on this threshold, unless it is changed to very large values (> 1.0e-6). Then, the contrail ice becomes mostly" invisible" to the radiation scheme. Thus, only modification in the natural ice class is present.*
*We follow your suggestion and add after page 5, line 28:*
The same threshold value is used in Seifert and Beheng (2006) for grid-scale natural ice clouds.

**11. Page 5, lines 30 ff. The team around Ping Yang has developed an ice particle parameterization including smaller ice particles in recent years (see Bi and Yang, 2017, e.g.). The existence of such parameterizations should be mentioned and such new parameterizations could be used at least in future studies.**
*We are aware of this work and other (e. g. Fu et al. (2007)). Implementing a new description of cloud optical properties is a rather large effort, as much of the tuning of the COSMO model depends on the results of this part of the code. Currently, we are working on the cloud optical properties but this work will not be finished and ready to use in the near future.*
*We add the following after page 7, line 5:*
Other parameterizations exist that can compute reliable values for optical properties of small ice crystals with sizes down to 0.2 µm (Bi and Yang, 2017). For future studies, using such an approach clearly could overcome the necessity of the threshold described above.

**12. Page 9, line 2: The introduction to this section, with "In contrast to previously mentioned global modeling studies, ..." and with "globally averaged fuel consumption …" is no longer true if you mention CoCiP properly.**
We change:
"In contrast to previously mentioned global modeling studies…"
*to*
In contrast to most of the previously mentioned global modeling studies

**13. Page 7, line 4: replace "a potential function" by "a power law" or similar.**
*We change:*
"a potential function"
*to*
power law

**14. Page 7, line 13: "Microphysical properties" – this is a very vague term. What do you mean? If you mean optical extinction, I am not sure that your statement is correct (I expect that extinction and optical depth are equally sensitive to number and mass).**
*At a given time, certainly both, mass and number control optical properties. What we wanted to say is that optical properties of AGED contrails depend on the initial ice crystal number and not much on the initial ice mass. We expand the description to avoid misunderstandings.*
*We change:*
*"Microphysical properties of contrails depend a lot more on the number of ice crystals than on the ice mass after the vortex phase (Unterstrasser and Gierens, 2010b)."*
*to:*

Microphysical properties of aged contrails depend a lot more on the number of ice crystals than on the ice mass after the vortex phase (Unterstrasser and Gierens, 2010b). The initial ice mass is of minor importance, as the later growth of contrail ice crystals and the related ice mass evolution in a persistent contrail is mainly controlled by the ambient water vapor supply .On the other hand, the ice crystal number changes only slowly in a long-living contrail. Hence, its initial value determines the typical ice crystal sizes in the evolving contrail-cirrus (for a given environmentally controlled ice mass), which affects the radiative properties and the sedimentation-related life cycle.

**15. Page 8, line 19: 600 m is a large upper bound which is reached very rarely. To be fair, the lower bound should be correspondingly small (100 m; even smaller values occur for small business jets).**
*We change the limits to 100 m and 500 m.*

**16. Page 9, line 2 ("In contrast to...") : Note that some previous global model studies used similar data for the whole year of 2006 ("ACCRI" waypoint data). Hence, this is not really a big step forward.**
**The word "exact" does not fit well to this description. When are data exact? Also "real time-based" data is not really the right term. I would simply say you use traffic waypoint data from transponder data (not radar).**
*We change:*
*Rather than statistical calculations for globally averaged fuel consumption, the basic data consist of exact flight trajectories over a limited area that are recorded from real time-based data (flightradar24.com, 2015).*
*To:*
Rather than statistical calculations for globally averaged fuel consumption, or radar data, the basic data consist of traffic waypoint information over a limited area recorded from transponders on the plane (flightradar24.com, 2015).

**17. Page 11, line 3: Here and at several later places you could also cite observation results. Here, e.g., Petzold et al. (1997) and Heymsfield et al. (1999) found strongest growth at the edges of contrails from in-situ measurements.**
**The discussion of observability of contrail cirrus is not needed in this paper. Otherwise, the statements are controversial and would require more in-depth discussion for completeness. For example, observations discriminate contrails and cirrus also based on the concentration of exhaust trace gases and aerosols and use trajectory analysis, partly in correlation with air traffic and meteorological situation history (Voigt et al., 2017). I suggest reducing this discussion in this paper. It is not needed for this paper.**
*We now mention the studies by Petzold and Heymsfield , additionally we included Poellot et al 1999 and Voigt et al 2017 at a later occasion.*
*We change page 11, line 10:*
*"Consistently, large-eddy simulation studies indicate the strongest growth at the edges of a contrail (Unterstrasser and Gierens, 2010a; Lewellen et al., 2014)."*
*to:*
Consistently, large-eddy simulation studies (Unterstrasser and Gierens, 2010a; Lewellen et al., 2014) and in-situ measurements (Petzold et al., 1997; Heymsfield et al., 1998) indicate the strongest growth at the edges of a contrail.

[revised manuscript text omitted]

**19. Page 14, line 11-12: Why did contrails form only between 11 km and 13 km altitude? Is this because there was no traffic below and above, or because of drier and warmer air above and below?**
*We add in page 17, line 6:*
This is caused by the absence of air traffic below this layer.

**20. Page 15, line 2-4: Again you compare to LES only. You could as well compare to observations. Such data are readily available from Schumann et al. (2017). If you would have plotted the ice particle concentration per volume along a line though the contrail cirrus clouds, you could have compared to the findings in, e.g., Voigt et al. (2017), Fig. 6.**
*We follow your suggestion and add the following figure and description:*

In Fig. 8, the relative occurrence of ice crystal number concentrations and temperature for the cross section shown in Fig. 7 is depicted. The relative occurrence is normalized with the sum over all values. Both, reference simulation (Fig. 8a) and aviation simulation (Fig. 8b) are similar for higher temperatures (i. e. lower heights) up to 220 K. For lower temperatures, high number concentrations up to 7 $cm^{-3}$ occur in the aviation simulation, whereas number concentrations clearly decrease strongly with temperature in the reference simulation. Here, a rough comparison to measurement data can be made. In Voigt et al. (2017), mid-latitude cirrus clouds and contrails where probed in-situ during an aircraft measurement campaign. Comparing their Fig. 6(b) to Fig. 8, a similar increase in n below temperatures of about 220 K is found. Therefore, most likely, the high values occurring in the aviation simulation and not in the reference simulation, are due to aviation induced clouds.

[Figure]

Figure 8. Relative occurrence of ice crystal number concentration versus temperature for natural cirrus, contrail and contrail cirrus at 12 UTC for the cross section along the black line in Fig. 4; a) reference simulation; b) aviation simulation.

*We rewrite section 4.3 and use other satellite images with much better resolution and more precise information about the algorithm.*

*We replace Fig 8 (now Fig. 10) with the following:*

[Figure]

Figure 10. Top row: satellite image (MODIS True Color - Corrected Reflectance) (NASA/GSFC/ESDIS, 2018); center row: optical depth at 1.115 µm of all ice clouds for the aviation simulation; bottom row: optical depth of all ice clouds for the reference simulation; left column: 10 UTC; right column: 12 UTC.

In the following, satellite images (created with Global Imagery Browse Services (GIBS) NASA/GSFC/ESDIS, 2018) are shown in Fig. 10 for a qualitative assessment of the simulations. The panels a and b show the "MODIS Terra Corrected Reflectance True Color" at 10 UTC and 12 UTC for the simulated day, respectively, both with a resolution of 250 m. The "True Color" composition consists of MODIS bands 1, 4 and 3 (NASA/GSFC/ESDIS, 2018). Beside a cloud bank over the North and Baltic Seas and fog over Southern and Western Germany, a considerable number of line-shaped contrails and diffuse cirrus clouds are present across both satellite images. Main contrail clusters are found over Central Germany for both situations. Contrails can also be identified over the Netherlands and Belgium, over the Czech Republic, and south of the Alps. At 12 UTC, the rather widespread high-level cloud cover seems to have decreased compared to 10 UTC.

Comparing Fig. 10a and Fig. 10e, obviously, the reference simulation underestimates the coverage of high clouds in the center of the domain, which, in large parts, consists of contrails and contrail cirrus Fig. 10c. Clearly, the amount of cloud cover seems to be underestimated also in the aviation simulation at 10 UTC. This discrepancy is probably due to the fact that air traffic was switched on at 08 UTC and earlier flight movements are disregarded in our simulation. Hence, the simulation evaluation at 10 UTC neglects all contrails older than 2 hours. The comparison at 12 UTC is more favorable and observations match much better with the aviation simulation than with the reference simulation.

…

Acknowledgements:
We acknowledge the use of imagery from the Land Atmosphere Near real-time Capability for EOS (LANCE) system and services from the Global Imagery Browse Services (GIBS), both operated by the NASA/GSFC/Earth Science Data and Information System (ESDIS, \url{http://earthdata.nasa.gov}) with funding provided by NASA/HQ.

**23. Page 19, Fig. 11: It took me some time to find the red circle. Please add the coordinates (12°E, 53°N) in the figure caption.**
*Done*

**24. Page 22, line 11: What is a "nominal capacity"?**
*The nominal capacity (or nominal power) is the power that an electrical device can produce (or handle) when operating in a reasonable manner as proposed by the manufacturer.*
*We change the paragraph from:*
*A nominal capacity is assumed, consisting of a specific PV module and PV inverter combination. The tilt and orientation of the PV module as well as the technical specifications are taken from Rieger et al. (2017).*
*to:*
For a specific combination of a PV module and a PV inverter combination, a nominal power and a reasonable tilt is assumed. These technical specifications are taken from Rieger et al. (2017).

**25. Page 22: Section 4.5: This section depends on the accuracy of the model used (with about 2 km horizontal and 300 m (or 400 m at the tropopause?) vertical resolution) since the early contrail ice crystal loss certainly depends on the time scale of plume mixing. This should be mentioned.**
*The early contrail ice crystal loss is part of the initialization in COMSO-ART which is based on LES results. By setting lambda_Ns = 1 we simply disregard any losses that appear in a young contrail. This sensitivity test is not affected by COSMO grid resolution and plume mixing within COSMO.*

**26. Page 23 line 10-12. The numbers given depend on temperature. See Fig. 5 in Schumann et al. (2017) and Fig. 6 in Voigt et al. (2017), and many other related studies. In fact, this should have been discussed earlier in the text. In the conclusion, it should be said that the numbers for IWC and other contrail-cirrus properties are valid for the specific meteorological situation considered.**

*We add in page 23, line 10:*
For a single case study, it was shown,  …

**Additional references:**

Bi, L., and P. Yang: Improved ice particle optical property simulations in the ultraviolet to far-infrared regime, J. Quant. Spectrosc. Radiat. Transf., 189, 228-237,
doi:10.1016/j.jqsrt.2016.12.007, 2017.

Caiazzo, F., Agarwal, A., Speth, R. L., and Barrett, S. R. H.: Impact of biofuels on contrail warming,
Environ. Res. Lett., 12, ,
doi:10.1088/1748-9326/aa893b, 2017.

Duda, D. P., Minnis, P., Nyuyen, L., and Palikonda, R.: A case study of the development of contrail clusters over the Great Lakes, J. Atmos. Sci., 61, 1132-1146, 2004.

Heymsfield, A. J., Lawson, R. P., and Sachse, G. W.: Growth of ice crystals in a precipitating contrail,
Geophys. Res. Lett., 25, 1335-1338,
doi: 10.1029/98GL00189, 1998.

Iwabuchi, H., Yang, P., Liou, K. N., and Minnis, P.: Physical and optical properties of persistent contrails: Climatology and interpretation, J. Geophys. Res., 117, D06215,
doi:10.1029/2011JD017020, 2012.

Petzold, A., Busen, R., Schröder, F. P., Baumann, R., Kuhn, M., Ström, J., Hagen, D. E., Whitefield, P. D., Baumgardner, D., Arnold, F., Borrmann, S., and Schumann, U.: Near-field measurements on contrail properties from fuels with different sulfur content, J. Geophys. Res., 102, 29867-29880,
doi: 10.1029/97JD02209, 1997.

Schumann, U., Penner, J. E., Chen, Y., Zhou, C., and Graf, K.: Dehydration effects from contrails in a coupled contrail-climate model, Atmos. Chem. Phys., 15, 11179-11199,
doi:10.5194/acp-15-11179-2015, 2015.

Schumann, U., Baumann, R., Baumgardner, D., Bedka, S. T., Duda, D. P., Freudenthaler, V., Gayet, J.-F., Heymsfield, A. J., Minnis, P., Quante, M., Raschke, E., Schlager, H., Vázquez-Navarro, M., Voigt, C., and Wang, Z.: Properties of individual contrails: A compilation of observations and some comparisons, Atmos. Chem. Phys., 17, 403-438,
doi:10.5194/acp-17-403-2017, 2017.

Schumann, U., and Heymsfield, A.: On the lifecycle of individual contrails and contrail cirrus,
Meteor. Monogr., 58, 3.1-3.24,
doi: 10.1175/AMSMONOGRAPHS-D-16-0005.1, 2017.

Vázquez-Navarro, M., Mannstein, H., and Kox, S.: Contrail life cycle and properties from 1 year of MSG/SEVIRI rapid-scan images, Atmos. Chem. Phys., 15, 8739-8749,
doi:10.5194/acp-15-8739-2015, 2015.

---

## Author Comment (AC2) · 29 Jan 2018

**Dear Referee 2,**

We thank a lot for your valuable comments and suggestions. We followed them as explained below.

The reviewers comments are repeated in **bold letters**, our replies are given in *italic*, and text modified or added to the manuscript is given in blue.

The authors use a regional scale model and contrail parameterization to simulate contrails and cirrus clouds occurring over central Europe during a single day – December 3rd, 2013. The simulated cloud cover and ice crystal mass mixing ratios are used in an online radiation scheme to understand the impact of aviation on direct and diffuse shortwave radiation reaching the surface, which in turn, affect the production of photovoltaic power. Overall, it is reported that aviation-induced cloudiness reduces PV power production by up to 10%. Assumptions related to the emissions index of ice crystals and crystal loss during the contrail vortex phase significantly alter this estimate. This is an interesting case study, which should be worth publishing in ACP; however, the limited time and spatial coverage of the reported model simulations limit the usefulness of authors' conclusions for broader understanding the relevance of aviation contrail cirrus to solar energy production. It certainly would be nice to see more data points for other spatial locations or time periods (e.g., summer). The following comments must be adequately addressed before I can recommend that this paper be acceptable for publication.

Thanks a lot for your review and your interest in our work. We see (and share) your opinion of the problem concerning the limited time and spatial coverage. However, this study should be seen only as a first step: developing the model, doing some comparison with observations and LES. Future studies with longer simulation time and other (larger) domains can follow this work.

A certain problem, at least with the presented setup is the following: The regional model needs lateral boundary data. From global models, we can get those information for the meteorology (partly also for gas phase and aerosols). However, neither data on contrails or contrail cirrus nor on aviationmodified cirrus clouds is available currently. Those would be needed to drive a model producing realistic results for longer time. Currently, several efforts to implement contrails into the ICON model are undertaken. Using this will overcome this issue.

Indeed, one day of simulation is not enough for providing a significant and statistically robust signal. We improved the figure that shows the time series of radiation related quantities by adding standard deviations as error bars.Furthermore, we additionally compare values for the mean for the entire simulation domain to the ones obtained for the selected area.

*Further studies will follow, applying the model to other days, time of year, regions etc. to further investigate and enhance the significance (and relevance) of the influence of aviation on PV production.*

**1. Pg. 2, Line 16-17: This sentence is confusing. What is being claimed here – that this is the first time a regional scale model has been applied to study contrails and contrail cirrus? I don't think a statement like this is really necessary, but in any case, please be specific with what is being claimed as novel.**

We delete the sentence in page 2, line 16-17:

[revised manuscript text omitted]

**4. Pg. 3, Line 20: In the following what? This sentence is confusing.**

We change in page 3, line 20: "In the following, ..." to: In this section, ...

**5. Pg. 4, Line 21: How often and under what conditions are these limits actually reached in the simulations?**

In general, critical processes are sedimentation and melting / sublimation of hydrometeors. To assure numerical stability, the limits are introduced within a clipping routine at the end of the microphysics. As mass and number concentrations are treated separately, it may occur, that at the end of a time step, e. g. due to sedimentation, a rather large value for the number concentration remains with a small amount of mass present or vice versa. This would result in unphysically small or large crystals. The same may occur during melting / sublimation. Here, in the first place, only the mass concentration is changed resulting in very small mean masses for the ice crystals.

Apart from this, the microphysical scheme is designed in a way to avoid reaching these limits. E. g. ice gets converted to snow via accretion or graupel starts melting when getting too large by simply sedimenting in lower and warmer layers.

About the "how often":

Tab.1 Occurrence of limits violated for both classes and simulations, each for simulation hour 12

|                      | aviation     |                | reference    |
|----------------------|--------------|----------------|--------------|
|                      | cirrus class | contrail class | cirrus class |
| m < m MIN | 0.002 %      | 0.005 %        | 0.001 %      |
| m> m MAX  | 0.03 %       | -              | 0.03 %       |

In Tab.1, the relative occurrence of violated limits is shown for cloudy grid boxes both simulations and the classes considered for the 12th hour simulated. Note that only cloudy grid boxes (i. e. mass concentration > 0) are considered. Apparently, the limits for both classes are reached only rarely, also when including contrail ice.

For the case study shown, also the plots for effective radii (in Fig. 3, 4, 6, 7) may give a hint: Limiting the number concentration of ice crystals to assure a fixed mean mass (m) would result in a constant value for the effective radius ( $r_e = r_e(m)$ ). As the respective plots show in large parts at least some variability over most areas for values of  $r_e$ , the limits most likely are not reached. 6. Pg. 5, Line 26-28: How are contrail and contrail cirrus ice distinguished from cirrus ice given the statement on Pg. 5, Line 5 that the cirrus and contrail cirrus ice classes are lumped together?

As mentioned in page 5, line 5, we cannot distinguish directly between contrail cirrus und natural cirrus directly, as soon as the contrail cirrus gets transferred to the "natural" cirrus class. The statement in page 5, line 26 – 28 actually refers only to the fact, that the contrail ice class is

represented in the radiation algorithm with its own cloud cover and optical properties.

*The contribution to the natural cirrus class can only be seen comparing to the reference simulation. We change:*

"To include contrails and contrail cirrus into the radiative algorithm, we include a contrail ice cloud cover determined from the contrail and contrail cirrus ice mass mixing ratio." to:

To include contrails and contrail cirrus in the radiative algorithm, we include a contrail ice cloud cover determined from the contrail ice class mass mixing ratio.

...

As mentioned before, the aviation contribution to the natural cirrus ice class can only be determined by comparison with the reference simulation.

**7. Pg. 7, Line 26: Is there a citation for the assumed EI\_iceno?**

We add in page 7, line 26: ... following Unterstrasser, 2014.

**8. Pg. 8, Line 18: What is the vertical resolution of the model?**

We change in page 8, line 18:

"In the vertical direction, it is reasonable to assume that ice crystals are distributed over the whole grid layer."

to:

In the vertical direction, it is assumed that ice crystals are distributed over the whole grid layer. Close to the ground, the vertical grid size is about 10 m and increases to 300 m at the tropopause.

**9. Pg. 10, Line 30: Is this sentence referring to Figure 3 instead of Figure 4?**

*This is true. We move it to the paragraph above.*

**10. Pg. 12, Line 3-4: What properties are being referred to here? I certainly wouldn't say that the number concentration in 4e and 4f are similar, and there are also large differences in IWC in 4b and 4c.**

Changed:

"Again, the properties of the aged contrails and the natural cirrus around Germany are similar." to:

Here, local enhancements in both IWC and n occur in the cirrus ice class, where aged contrails are transferred to (Fig. 4e), compared to the reference simulation (Fig. 4f).

**11. Pg. 14, Lines 11-12: Why do contrails only form at altitudes between 11 km and 13 km? Is this because air traffic is restricted to these altitudes on this line or are there lower altitude flights but the Schmidt-Appleman criterion is only satisfied at these altitudes?**

Taking into account the rather high values for RHi also below 11 km, the Schmidt-Appleman criterion likely would be fulfilled.

But the traffic data we use hardly contains flights taking place below 11 km expect for climbing and descending. At least for the cross section selected, apparently none of these movements was taking place...

This is caused by the absence of air traffic below this layer.

12. Pg. 15, Line 6: Should the first word be "below"?

Changed "beyond" to below

13. Pg. 15, Line 10: Does the model account for downward subsidence of the aircraft vortices and plumes or is the vertical structure in the modeled contrails only due to gravitational settling of the larger ice crystals? The enhancements in ice number shown in Figure 6 appear to occur at flight level, but the enhancement in IWC is below flight level.

Change paragraph starting in page 15, line 5:

"In the aviation simulation, changes in the natural ice class can be found mainly at heights where contrails form and slightly beyond. Here, local increases in IWC and N occur. Keeping in mind the simple design of our model setup, those ice crystals represent both fall streaks of contrails as well as the transition into contrail cirrus."

to:

In the aviation simulation, changes in the natural ice class can be found mainly at heights where contrails form and slightly below of it. The enhancement of ice number concentrations occurs mostly at flight levels, whereas an increase in \$IWC\$ is found below. During the initialization, contrail ice crystals are vertically distributed over the whole grid layer and this indirectly accounts for the initial wake vortex induced vertical expansion of a contrail. Within our simulation, the vertical structure of the contrails is determined only due to the gravitational settling of the larger ice crystals.

14. I would like to see the satellite observations more directly integrated into the discussion surrounding Figures 3-4 rather than in its own section since I think that it can provide a lot of context and validation for the model results. Figures 8 and 9 as they stand now are kind of on their own and not particularly informative other than to denote that there are thin, high-level cirrus and no low clouds. The cirrus clouds shown in Figure 8a appear to be much more diffuse than the MODIS imagery for this time period, with the MODIS images showing a lot of contrail structure and providing a good snapshot of the time evolution of the scene during the two simulations. I suggest the authors strike Figures 8 and 9, and add MODIS satellite images at 1000Z and 1150Z either as part of Figures 3 and 4 or as a separate figure before them. Such an example figure created from worldview.earthdata.nasa.gov images is on the next page with detailed web references at the end of this review.

We have rewritten parts of section 4.3 (Comparison with Satellite Observation). We use the great source for images you suggest for both time steps considered. However, we refrain from including the discussion into the section before. There, we want to focus on the behavior of the model itself and consequences that arise thereof. A future study for situations where preferably in-situ measurements are available will then also include a more detailed verification and comparison with satellite data.

---

## Author Comment (AC3) · 29 Jan 2018

Dear Referee 3,

We thank a lot for your valuable comments and suggestions. We followed them as explained below.

The reviewers comments are repeated in **bold letters,** our replies are given in *italic,* and text modified or added to the manuscript is given in blue.

**This paper describes a parametrization of contrails that is embedded in the two moment cloud microphysics of the COSMO regional atmospheric model. It is followed by a case study including the impact of contrails and contrails-cirrus on short wave radiation and on possible PV production. The paper presents interesting new results, but before being published the authors should address the following points.**

*Thanks a lot for your remarks. Most of your points address errors and mistakes that one does only recognize when having a very close look at the manuscript. Your review helped a lot improving our manuscript.*

**General remarks:**

**1. I was a bit surprised that the authors have chosen a model with a complete representation of atmospheric chemistry and aerosols (COSMO-ART) for this study. From the description of the parametrization in Section 2 it seems to me that changes by aircraft to atmospheric chemistry as well as soot emissions are not directly considered. Thus it seems that the usage of the COSMO model (without the computationally expensive ART modules) in conjunction with the 2-moment scheme of Seifert and Beheng (2006) are sufficient for the present study. If this is not the case it should be shown more clearly why the ART modules are needed.**

*Indeed, no interactions of aircraft emissions are considered in this study. Also, no soot emissions are taken into account explicitly.*

*But COSMO-ART comprises more than only parameterizations for aerosol dynamics and gas phase chemistry. For example, the infrastructure, e. g. the ART-tracer structure and modules to read emission data are used for the contrails parameterizations. Also the coupling of aerosol-dynamics and the microphysics is adapted to calculate contrail microphysics.*

*In the future, we want to extend the model to also consider aircraft emissions. Therefore, the entire COSMO-ART model is referred to.*

*We add in Sec. 2 (Model Description) after*

*"In this section, the parameterizations to calculate the microphysical properties of ice crystals and the modifications to represent contrails are presented."*

The model system COSMO-ART comprises a detailed treatment of aerosol dynamics and gas-phase chemistry (Vogel et al., 2009). However, most of these features are not used for the simulations shown in this study. Nevertheless, large parts of the infrastructure contained in COSMO-ART, e. g. the tracer structure and modules for reading emission data are adopted for the contrails parameterizations.

**2. Whereas the case study in section 4 is well described, sections 2 and 3 are rather hard to read and understand and would profit from a rewrite. In section 2.1 it is very difficult to understand where the description of the standard cirrus class is ended and the description of the newly introduced contrail class starts. I would thus recommend separating by introducing a new section. Also this section would highly profit from a table explaining the differences between both classes as well as a schematics such as Fig 1 in Salzmann et al. (2010) to highlight the interactions of the newly introduced contrail ice class with the other ice classes. A Table is provided at the end of the paper explaining the differences, but is not referenced in the text. I would moreover suggest in Table 1 to distinguish more clearly between the cirrus and contrail ice class. Similarly in section 2.2 a Table would help to understand the differences between the interactions with radiation of both classes.**

*Refrain from presenting schematics, as very complex for SB06, also actually contrail – cirrus interaction not too complicated to understand.*
*We change Tab. 1 to more clearly distinguish between contrail ice class and natural ice class.*
*We change in section 2.1 The Contrail Ice Class, page 4, line 4:*
*"A longer description including various formulae is deferred to the appendix."*
*to:*
A longer description including various formulae as well as a table showing the different coefficients characterizing both the contrail and the cirrus ice class (Tab. 1) is deferred to the appendix.
*Section 2.2: There are no differences in the interaction of contrail / natural ice with radiation.*

**3. It seems quite strange that a completely new scenario "bio fuels" is introduced in the last section of the paper, but referenced already in Fig 12. I would suggest to keep section 4.5 as sensitivity case study, and not introduce a new scenario here. As shown by Ferrone (2011) biofuels also have an impact on the Appleman-Schmidt criterion and this would need to be changed accordingly.**
*We do not aim at introducing a new "bio fuel" scenario where also effects on contrail formation had to be included (as you correctly mention). It is common practice in contrail studies to study the sensitivity to the initial ice crystal number. Various reasons (uncertainty and variability of EI_iceno-value, biofuels effect) to do so exist.*

**Specific remarks:**
**4. In the last line of page 2, the resolution of Global Circulation Models (GCMs, the abbreviation is not introduced) are given as 250km, however most recent models have a resolution of 50 km or higher (IPCC, 2015).**
*This is a fair point, but the argument is still valid comparing to 50 km resolution. We change in page 2, line 32:*
*"Compared to GCM parameterizations, this omission seems acceptable in a regional model, as the spatial resolution is much higher (horizontal grid size of 2.8 km 250 km in a GCM) ."*
*to*
Compared to GCM (Global Circulation Model) parameterizations, this omission seems acceptable in a regional model, as the spatial resolution is much higher (horizontal grid size of 2.8 km versus 50 km in a GCM).

**5. Caption of Figure 2: The abbreviation COSMO-DE is not introduced**
*Actually, the name of the domain is not important. COSMO-DE is the domain, used by the German Weather Service to run their high resolved forecast for Germany*
*We change the caption of Fig. 2 from*
*"Flight trajectories for the COSMO-DE domain …"*
Flight trajectories for the simulated domain …

**6. Figures 3, 4, 6 and 7 would become more interesting and easy to interpret if a difference plot between the middle and right column would be added.**
*We avoid difference plots here, as their interpretation becomes problematic. Due to the design of our model, we cannot distinguish between contrail cirrus and natural cirrus in the cirrus ice class.*
*By formation of contrails in the aviation simulation, humidity changes and thus partly, no cirrus clouds form, where they do in the reference simulation. Also the properties of natural cirrus may change in the aviation simulation.*

**7. Figure 8 and 9: If I understood correctly, the black boxes should highlight the same areas but they are slightly shifted.**
*Indeed, the boxes were not covering the same areas, thanks for the remark. Due to restructuring the section, Fig. 9 (black boxes) are not shown anymore.*

**8. In the list of abbreviations given on page 31 and 32 some Units are erroneous. Units such as kgˆ(-/mu), mˆ(-/mu); kgˆ(-b_vel); kg ˆk do not exists.**

*In case, one insists on SI units, this is a fair point, although the quantities are defined as such in Seifert and Beheng (2006).*

*We convert the equations, where the strange units occur (A6, A7) into numerical value equations and remove the units.*

*We add in Appendix A:*

*page 24, line 6:*

Here, m is in units of kg.

*page 25, line 5:*

In Eqs. A6 and A7, m and  \overline{m} are in kg;  and v and \overline{v}_{k} are in in m s-1.

*page 25, line 20:*

… with, L in units of m:

*Also Tab 1 and the list of abbreviations is changed accordingly.*

**References:**

**Ferrone, A, (2011): Aviation and climate change in Europe: from regional climate modelling to policy-options.**
**http://hdl.handle.net/2078.1/74779**

**IPCC, 2013: Climate Change 2013: The Physical Science Basis. Contribution of Working Group I to the Fifth Assessment Report of the Intergovernmental Panel on Climate Change [Stocker, T.F., D. Qin, G.-K. Plattner, M. Tignor, S.K. Allen, J. Boschung, A. Nauels, Y. Xia, V. Bex and P.M. Midgley (eds.)]. Cambridge University Press, Cambridge, United Kingdom and New York, NY, USA, 1535 pp, doi:10.1017/CBO9781107415324.**

**Salzmann, M., Y. Ming, J. C. Golaz, P. A. Ginoux, H. Morrison, A. Gettelman, M. Krämer, and L. J. Donner, 2010: Two-moment bulk stratiform cloud microphysics in the GFDL AM3 GCM: Description, evaluation and sensitivity tests. Atmospheric Chemistry and Physics, 10, 8037-8064, doi:10.5194/acpd-10-6375-2010.**

**Seifert, A. and Beheng, K. D.: A two-moment cloud microphysics parameterization for mixed-phase clouds. Part 1: Model description, Meteorol. Atmos. Phys., 92, 45 – 66, doi:10.1007/s00703-005-0112-4, 2006.**

---

## Author Response (AR1)

Dear Mr. Schumann,

We thank a lot for your valuable comments and suggestions. We followed them as explained below.

The reviewers comments are repeated in **bold letters**, our replies are given in *italic*, and text modified or added to the manuscript is given in blue.

This is a nice model study of contrail and contrail cirrus formation on a regional scale. The approach is straightforward in extending a two-moment cloud microphysics scheme by including a separate contrail ice class. The scale jump from young contrails at aircraft wake vortex scales to grid scales is approximated by assuming that the contrail ice spreads immediately over a grid scale (vertically and horizontally). Part of the effects of this strong simplification is corrected by using an ice-crystal loss parametrization derived from LES results.

Of course this simple approach is possible only because the model study is restricted to regional scales, with just one day of simulation (actually only 8 h of contrail formation). The model is indeed useful to study regional effects of contrail cirrus on shortwave surface radiation and possibly on weather prediction. If applied for climate studies at global scales and for long simulation periods (years), the present approach would suffer from the same problems as other global models. For example, a regional model would require contrail cirrus boundary conditions if run for longer time periods, which are nontrivial if contrail effects outside the region impact the meteorology at inflow.

Any climate simulation would also require coupling to oceans etc. Nevertheless, the approach is useful for regional studies and interesting.

Of course, finally, the study should be published, though several minor and some major text issues need to be considered, as listed below, before the paper is acceptable.

The authors present a new contrail cirrus parameterization within the COSMO-ART model and study the impact of contrail cirrus on the SW radiation. The work is different to existing studies since they implemented their parameterization in a weather forecasting model. This would actually enable the authors to compare their results to observations but they do not do that. Instead they discuss the development t of a contrail field in simulations only. Nevertheless, I find the model itself and the impact on the SW radiation interesting. There are a couple of major issues that need clarification and/or sorting out. The paper needs to be expanded regarding a more detailed evaluation with observations and a better comparison with earlier work. *Dear Mr. Schumann,*

Thanks a lot for your interest in our work and your detailed review. The issues raised in this review as well as the many questions and explanations clearly help to improve this manuscript. We hope to have addressed them to a satisfactory extent.

**1.** Page 1, line 11: insert "and humid" after "hot". *Done*

2. Page 1, line 13: Note that the threshold temperature is pressure dependent; hence, -45°C is perhaps even too rough. More relevant is the fact that below a temperature near -38°C to -40°C, contrail particles, which are formed in liquid phase initially, freeze homogeneously and quickly to form ice particles which then persist in air with relative humidity below liquid saturation (but above ice saturation).

We change in page 1, line 13:

Following the Schmidt-Appleman-Criterion (Schmidt, 1941; Appleman, 1953; Schumann, 1996), contrails form only when the ambient temperature is below a threshold temperature of roughly - 45°C. With a favorable state of the atmosphere, i.e. characterized by supersaturation with respect to ice, the originally line-shaped contrails undergo various physical processes at the micro scale...

to:

Contrails form in case, the Schmidt-Appleman-Criterion (Schmidt, 1941; Appleman, 1953; Schumann, 1996) is fulfilled, i.e. the ambient temperature is below a threshold of around -45°C. With plume temperatures near -38°C to -40°C, contrail particles, which are formed in the liquid phase initially, freeze homogeneously and quickly to form ice crystals. Those ice crystals grow in air with relative humidity above ice saturation and contrails persist. With such a favorable state of the atmosphere, the originally line-shaped contrails undergo various physical processes at the micro scale ...

**3.** Page 1, line 17: There are other long-life-time contrail observations, but I agree, the one described by Minnis et al. (1998) is an early example. Others were summarized in Schumann and Heymsfield (2017), who also review the definition of contrail cirrus and other related knowledge. *We add in page 1, line 17:*

Other examples of long-lifetime contrail observation are summarized by Schumann and Heymsfield (2017).

4. Page 1, line 20: It is not clear whenever important properties are "sufficiently investigated", and there are many more observation (see Iwabuchi et al., 2012; Vazquez-Navarro et al., 2015; Schumann et al., 2017) and model studies than those covered in the IPCC report (Boucher et al., 2013).

*Even taking into account the newer observations you mention we still believe that our statement is reasonable.*

5. Page 2, line 17-18: "whether this study is the first of its kind" is at least debatable. In particular since the authors do not discuss earlier attempts like those of Duda et al. (2004) at this place. But I agree that the simplicity of the present approach including a balanced microphysics model (similar to recent approaches in global models) is attractive and the authors can claim a fresh approach. We weaken the statement and mention the study of Duda later in the manuscript as it also uses real flight data. However, we do not consider the contrail advection tool of Duda to be a "full" contrail model, in the sense that ice microphysics is included. Hence, our aforementioned claim is not invalidated by disregarding the Duda work.

We add in page 2, line 18:

Another approach using commercial flight data to study contrails on a regional scale is described in Duda et al. (2004). Here, a combination of commercial flight data and coincident meteorological satellite remote sensing data was used to perform a case study of a widespread contrail cluster.

**Reference:**

Duda, D. P., Minnis, P., Nyuyen, L., and Palikonda, R.: A case study of the development of contrail clusters over the Great Lakes, J. Atmos. Sci., 61, 1132 – 1146, 2004.

6. Page 3, line 1-3: The method developed by Schumann (2012) and Schumann et al. (2015) (added reference, see below), though certainly with limitations, is still likely the only one covering the scale transition from thousands of single contrails to multi-year global climate cases. This method is not represented fairly in this citation (and in this introduction) which correctly mentions a problem but misses to mention the advantages of that approach. In fact, the mixed Lagrangian-Eulerian approach could be listed as a basic alternative to the Eulerian grid scale models. This approach has also been used by Caiazzo et al. (2017).

We changed the text accordingly:

There, a mixed Lagrangian-Eulerian approach is used instead of the usual Eulerian treatment. This approach allows covering the scale ranging from thousands of single contrails to multi-year global climate simulations (Schumann (2012); Schumann et al. (2015); Caiazzo et al. (2017)).

7. Page 3, lines 13-15: The paper seems to make a big deal out of using 8 h of traffic movements. There were earlier studies doing far more in that direction (e.g., Schumann, 2012; Voigt et al., 2017).

The authors did not consider a case for which in situ- and satellite observations and other model studies are available, such as the ML-CIRRUS observations of 10 April 2014 over Germany (Voigt et al., 2017), for which the waypoint-traffic data (partly also from flightradar24.com) are available for about 4 weeks and for nearly the whole of Europe. See, e. g., Fig. 4 in Voigt et al. (2017). The existence of such data for future studies should be mentioned.

Perhaps, the introduction should mention the use of satellite data. But it should mention that there were many studies of satellite data in the past (from polar orbiting and geostationary satellites) and also a large variety of in-situ observations is available. So far, I feel that the discussion in this paper is too much biased to LES results instead.

We now cite several more studies of contrail observations.

We do not think that the presentation or plausibility issues are overly biased towards LES, but it is clear that LES results build an integral part of the contrail initialisation used in our model.

At the time the manuscript was prepared, ML-Cirrus results were not published yet and it is not clear whether we would have been granted access at that phase to all the data needed to make conclusive comparisons.

We add the following after page 3, line 15:

In the future, further case studies should be performed for which in-situ observations of natural ice clouds and especially contrails are available.

*Fig. 1: 2014/04/10: Comparison of COSMO-ART ice crystal number concentration (contour) to SID3 measurements (ML-CIRRUS).*

We also performed a case study for 10 April, 2014 to compare with ML-CIRRUS data (Fig.1). In the future, we want to extend this work and present it in a separate study.

**8. Page 4, lines 20 to next page: The text explaining the parameters used in Eq. (2) appears a bit lengthy. I assume it can be reordered and shortened.**

We agree that the mentioned section is more technical than others. We condensed this part and moved the technical issues to the appendix.

9. Page 5, line 23: reorder "several input" -> "input several" Done

**10. Page 5, line 28: Is there any physical argument for using this maximum mixing ratio limit value? If not, say that this is arbitrarily taken. How sensitive are the results to this threshold?**

In fact, this threshold is used for grid-scale ice clouds in the radiation scheme. For reasons of consistency (and as we find no other suitable study to justify another number) we leave it as it is. Indeed, the results do not seem to depend strongly on this threshold, unless it is changed to very large values (> 1.0e-6). Then, the contrail ice becomes mostly" invisible" to the radiation scheme. Thus, only modification in the natural ice class is present.

*We follow your suggestion and add after page 5, line 28:* The same threshold value is used in Seifert and Beheng (2006) for grid-scale natural ice clouds.

11. Page 5, lines 30 ff. The team around Ping Yang has developed an ice particle parameterization including smaller ice particles in recent years (see Bi and Yang, 2017, e.g.). The existence of such parameterizations should be mentioned and such new parameterizations could be used at least in future studies.

We are aware of this work and other (e. g. Fu et al. (2007)). Implementing a new description of cloud optical properties is a rather large effort, as much of the tuning of the COSMO model depends on the results of this part of the code. Currently, we are working on the cloud optical properties but this work will not be finished and ready to use in the near future.

We add the following after page 7, line 5:

Other parameterizations exist that can compute reliable values for optical properties of small ice crystals with sizes down to 0.2  $\mu$ m (Bi and Yang, 2017). For future studies, using such an approach clearly could overcome the necessity of the threshold described above.

**12. Page 9, line 2: The introduction to this section, with "In contrast to previously mentioned global modeling studies, ..." and with "globally averaged fuel consumption ..." is no longer true if you mention CoCiP properly.**

We change:

"In contrast to previously mentioned global modeling studies..." to

In contrast to most of the previously mentioned global modeling studies

**13.** Page 7, line 4: replace "a potential function" by "a power law" or similar.**

We change: "a potential function" to power law

**14. Page 7, line 13: "Microphysical properties" – this is a very vague term. What do you mean? If you mean optical extinction, I am not sure that your statement is correct (I expect that extinction and optical depth are equally sensitive to number and mass).**

At a given time, certainly both, mass and number control optical properties. What we wanted to say is that optical properties of AGED contrails depend on the initial ice crystal number and not much on the initial ice mass. We expand the description to avoid misunderstandings. We change:

"Microphysical properties of contrails depend a lot more on the number of ice crystals than on the ice mass after the vortex phase (Unterstrasser and Gierens, 2010b)." to:

Microphysical properties of aged contrails depend a lot more on the number of ice crystals than on the ice mass after the vortex phase (Unterstrasser and Gierens, 2010b). The initial ice mass is of minor importance, as the later growth of contrail ice crystals and the related ice mass evolution in a persistent contrail is mainly controlled by the ambient water vapor supply .On the other hand, the ice crystal number changes only slowly in a long-living contrail. Hence, its initial value determines the typical ice crystal sizes in the evolving contrail-cirrus (for a given environmentally controlled ice mass), which affects the radiative properties and the sedimentation-related life cycle.

**15.** Page 8, line 19: 600 m is a large upper bound which is reached very rarely. To be fair, the lower bound should be correspondingly small (100 m; even smaller values occur for small business jets). *We change the limits to 100 m and 500 m.*

16. Page 9, line 2 ("In contrast to...") : Note that some previous global model studies used similar data for the whole year of 2006 ("ACCRI" waypoint data). Hence, this is not really a big step forward.

The word "exact" does not fit well to this description. When are data exact? Also "real time-based" data is not really the right term. I would simply say you use traffic waypoint data from transponder data (not radar).

We change:

Rather than statistical calculations for globally averaged fuel consumption, the basic data consist of exact flight trajectories over a limited area that are recorded from real time-based data (flightradar24.com, 2015).

То:

Rather than statistical calculations for globally averaged fuel consumption, or radar data, the basic data consist of traffic waypoint information over a limited area recorded from transponders on the plane (flightradar24.com, 2015).

17. Page 11, line 3: Here and at several later places you could also cite observation results. Here, e.g., Petzold et al. (1997) and Heymsfield et al. (1999) found strongest growth at the edges of contrails from in-situ measurements.

The discussion of observability of contrail cirrus is not needed in this paper. Otherwise, the statements are controversial and would require more in-depth discussion for completeness. For example, observations discriminate contrails and cirrus also based on the concentration of exhaust trace gases and aerosols and use trajectory analysis, partly in correlation with air traffic and meteorological situation history (Voigt et al., 2017). I suggest reducing this discussion in this paper. It is not needed for this paper.

We now mention the studies by Petzold and Heymsfield, additionally we included Poellot et al 1999 and Voigt et al 2017 at a later occasion.

We change page 11, line 10:

"Consistently, large-eddy simulation studies indicate the strongest growth at the edges of a contrail (Unterstrasser and Gierens, 2010a; Lewellen et al., 2014)."

to:

Consistently, large-eddy simulation studies (Unterstrasser and Gierens, 2010a; Lewellen et al., 2014) and in-situ measurements (Petzold et al., 1997; Heymsfield et al., 1998) indicate the strongest growth at the edges of a contrail.

**We change page 13, line 1:**

"The effective radii in the aviation simulation are typically one order of magnitude smaller than in natural cirrus clouds and do not exceed 10  $\mu$ m. These results are in reasonable agreement with large-eddy studies (Unterstrasser and Gierens, 2010a; Lewellen et al., 2014)." to:

The effective radii in the aviation simulation are typically one order of magnitude smaller than in natural cirrus clouds and do not exceed 10  $\mu$ m. These results are in agreement with large-eddy studies (Unterstrasser and Gierens, 2010a; Lewellen et al., 2014) and in situ observations (Poellot et al., 1999).

For comparison with Voigt et al. (2017), see answer to Comment 20.

We think that the paragraph on observability gives useful indications on how to interpret the COSMO results and hence we leave this section as is. The given reference (Unterstrasser et al, 2017) discusses this issue in more detail and the interested reader can read it.

A few thoughts on observability and your mentioned examples:

Lewellen et al (2014) contrast the spatial distribution of a passive chemical tracer and contrail-cirrus. Clearly, contrail area and plume area evolve differently over time, as ice crystals sediment as opposed to a passive tracer. Hence, exhaust trace gas measurements cannot be used to identify the complete contrail-cirrus. This method only allows tracking the contrail regions around the formation altitude where small ice crystals reside which have not yet fallen out of the aircraft plume.

**References:**

Heymsfield, A. J., Lawson, R. P., and Sachse, G. W.: Growth of ice crystals in a precipitating contrail, Geophys. Res. Lett., 25, 114 013, 1998. doi:10.1029/98GL00189

Lewellen, D., Meza, O., and Huebsch, W.; Persistent Contrails and Contrail Cirrus. Part I: Large-Eddy Simulations from Inception to Demise, J. Atmos. Sci., 71, 4399 – 4419, 2014. doi: 10.1175/JAS-D-13-0316.1

Petzold, A., Busen, R., Schröder, F. P., Baumann, R., Kuhn, M., Ström, J., Hagen, D. E., Whitefield, P. D., Baumgardner, D., Arnold, F., Borrmann, S., and Schumann, U.: Near-field measurements on contrail properties from fuels with different sulfur content, J. Geophys. Res., 102, 114 013 1997. doi:10.1029/97JD02209

Poellot, M. R., Arnott, W. P., and Hallett, J.: In situ observations of contrail microphysics and implications for their radiative impact, J. Geophys. Res., 104, 12 077–12 084, 1999. doi:10.1029/1999JD900109, http://dx.doi.org/10.1029/1999JD900109

18. Page 14, line 1 etc.: I agree that Fig. 5 exhibits a remarkably thick layer with apparently strong supersaturation (what is the maximum RHi value in this figure?). I wonder how this layer developed. Is this the results of initial conditions or the result so vertical lifting or radiative cooling? How realistic is the model result in this respect? As far as I understand, the high humidity coincides with some thin cirrus. How can the humidity persist so long in the presence of cirrus? If there is no cirrus yet then I would have expected some homogeneous ice nucleation at such high humidity values. Therefore: How realistic are the high RHi values? Please explain. *We changed the color coding as well as the scale in Fig. 5 . The upper bound is now 1.4 instead of 1.8. The maximum value for RHi is 1.34.*

---

## Author Response (AR2)

Dear Nikos,

We sincerely thank you for editing our manuscript and we also appreciate your comments and suggestions. We followed them as explained below.

The comments are repeated in **bold letters,** our replies are given in *italic,* and text modified or added to the manuscript is given in blue.

**1. Pg. 1, Ln. 12f: "Contrails form in case, the Schmidt-Appleman-Criterion (Schmidt, 1941; Appleman, 1953; Schumann, 1996) is fulfilled."**
**Delete comma**
*Done*

**2. Pg. 2, Ln. 2f: "Important properties, such as the optical depth or the spatial and temporal extent of occurrence, have not been sufficiently investigated and are also not quantified to a satisfactory extent (Boucher et al., 2013)."**
**Rephrase; refer to the studies reported by Schumann**
*Changed to:*
Although a number of observations (e. g. Iwabuchi et al., 2012; Vázquez-Navarro et al., 2015; Schumann et al., 2017) and other modeling studies have been performed, important properties, such as the optical depth or the spatial and temporal extent of occurrence, have not been sufficiently investigated and are also not quantified to a satisfactory extent (Boucher et al., 2013).

**3. Pg. 3, Ln. 22f :**
*Deleted a comment remaining from our own reviewing.*

**4. Pg. 4, Ln. 9f : "However, most of these features are not used for the simulations shown in this study. Nevertheless, large parts of the infrastructure contained in COSMO-ART, … are adopted …"**
**Add: "Although"; delete "Nevertheless"**
*Done*

**5. Pg. 26, Ln. 5: "For a single case study, it was shown …"**
**Specify which one**
*Changed to:*
Performing a case study for a single winter day over Central Europe, it was shown …

**6. Pg. 29, Ln. 13: "We appreciate the contructive comments…"**
**Typo**
*Changed to:*
We appreciate the constructive comments …

Dear Referee 3,

We thank you once again for your valuable comments. We followed it as explained below.

The reviewers comments are repeated in **bold letters,** our replies are given in *italic,* and text modified or added to the manuscript is given in blue.

**The authors have done a nice job responding to all of my comments, and I recommend that the manuscript be published after the following minor revision is made**

**1. I'm glad that it was felt that the addition of the MODIS imagery was helpful; however, from the revised Fig. 10, it appears to me that the imagery for 10 UTC and 12 UTC are reversed. From the orbital track times in Worldview, the MODIS Terra track overflies the region at roughly 10 UTC, while the MODIS Aqua track covers the region at roughly 12 UTC. The clouds observed by Terra are much more diffuse than the clouds observed by Aqua. The Terra orbit track also shows the SW->NE striping apparent on the left (west) edge of Figure 10b where the orbits are stitched together. Correcting this figure will also reverse the conclusions on Pg. 18, Line 29-30, and the subsequent discussion (Pg. 18, Ln. 31 - Pg. 20, Ln 7) should be modified to reflect that the satellite observations now support the model results - both satellite and model contrails and cirrus clouds increase from 10 UTC to 12 UTC, which is good.**

**Link to layered Worldview imagery is:**
**https://worldview.earthdata.nasa.gov/?p=geographic&l=VIIRS_SNPP_CorrectedReflectance_TrueColor(hidden),MODIS_Aqua_CorrectedReflectance_TrueColor,MODIS_Terra_CorrectedReflectance_TrueColor,Aqua_Orbit_Asc,Terra_Orbit_Dsc,Reference_Labels(hidden),Reference_Features(hidden),Coastlines&t=2013-12-03&z=3&v=-17.06324930328618,31.036661088764802,35.3750429177626,68.87135186462584**

*You are right, the figures were reversed. We switch Fig. 10a and Fig. 10b and change the discussion:*

[revised manuscript text omitted]

---

## Author Response (AR3)

Dear Nikos,

We sincerely thank you for editing our manuscript and we also appreciate your comments and suggestions. We followed them as explained below.

The comments are repeated in **bold letters,** our replies are given in *italic,* and text modified or added to the manuscript is given in blue.

**1.  Pg. 18, Ln. 32: "…consists of contrails and contrail cirrus Fig. 10c."**
*Changed to:*
… consists of contrails and contrail cirrus (Fig. 10c).

**2.  Pg. 18, Ln. 33f: "… the amount of cloud cover seems to be slightly underestimated also in the aviation simulation at 10 UTC."**
**Change "… also in the aviation simulation at 10 UTC." to "… in Fig. 10c."**
*Done*

**3. Pg. 20, Ln. 2f: "… 10 UCT"**
*Typo changed to:*
… 10 UTC

[revised manuscript text omitted]